# 7,8-Dihydroxyflavone is a direct inhibitor of human and murine pyridoxal phosphatase

Marian Brenner[1†], Christoph Zink[1†], Linda Witzinger[1†], Angelika Keller[1], Kerstin Hadamek[1], Sebastian Bothe[2], Martin Neuenschwander[3], Carmen Villmann[4], Jens Peter von Kries[3], Hermann Schindelin[2], Elisabeth Jeanclos[1]*, Antje Gohla[1]*

[1]Institute of Pharmacology and Toxicology, University of Würzburg, Würzburg, Germany; [2]Rudolf Virchow Center for Integrative and Translational Bioimaging, University of Würzburg, Würzburg, Germany; [3]Leibniz Forschungsinstitut für Molekulare Pharmakologie-FMP, Berlin, Germany; [4]Institute of Clinical Neurobiology, University Hospital, University of Würzburg, Würzburg, Germany

*For correspondence:
elisabeth.jeanclos@uni-wuerzburg.de (EJ);
antje.gohla@uni-wuerzburg.de (AG)

†These authors contributed equally to this work

**Abstract** Vitamin B6 deficiency has been linked to cognitive impairment in human brain disorders for decades. Still, the molecular mechanisms linking vitamin B6 to these pathologies remain poorly understood, and whether vitamin B6 supplementation improves cognition is unclear as well. Pyridoxal 5′-phosphate phosphatase (PDXP), an enzyme that controls levels of pyridoxal 5′-phosphate (PLP), the co-enzymatically active form of vitamin B6, may represent an alternative therapeutic entry point into vitamin B6-associated pathologies. However, pharmacological PDXP inhibitors to test this concept are lacking. We now identify a PDXP and age-dependent decline of PLP levels in the murine hippocampus that provides a rationale for the development of PDXP inhibitors. Using a combination of small-molecule screening, protein crystallography, and biolayer interferometry, we discover, visualize, and analyze 7,8-dihydroxyflavone (7,8-DHF) as a direct and potent PDXP inhibitor. 7,8-DHF binds and reversibly inhibits PDXP with low micromolar affinity and sub-micromolar potency. In mouse hippocampal neurons, 7,8-DHF increases PLP in a PDXP-dependent manner. These findings validate PDXP as a druggable target. Of note, 7,8-DHF is a well-studied molecule in brain disorder models, although its mechanism of action is actively debated. Our discovery of 7,8-DHF as a PDXP inhibitor offers novel mechanistic insights into the controversy surrounding 7,8-DHF-mediated effects in the brain.

## eLife assessment

Following small molecule screens, this study provides **convincing** evidence that 7,8 dihydroxyflavone (DHF) is a competitive inhibitor of pyridoxal phosphatase. These results are **important** since they offer an alternative mechanism for the effects of 7,8 dihdroxyflavone in cognitive improvement in several mouse models. This paper is also significant due to the interest in the phosphatases and neurodegeneration fields.

**eLife digest** Vitamin B6 is an important nutrient for optimal brain function, with deficiencies linked to impaired memory, learning and mood in various mental disorders. In older people, vitamin B6 deficiency is also associated with declining memory and dementia. Although this has been known for years, the precise role of vitamin B6 in these disorders and whether supplements can be used to treat or prevent them remained unclear.

This is partly because vitamin B6 is actually an umbrella term for a small number of very similar and interchangeable molecules. Only one of these is 'bioactive', meaning it has a biological role in cells. However, therapeutic strategies aimed at increasing only the bioactive form of vitamin B6 are lacking.

Previous work showed that disrupting the gene for an enzyme called pyridoxal phosphatase, which breaks down vitamin B6, improves memory and learning in mice. To investigate whether these effects could be mimicked by drug-like compounds, Brenner, Zink, Witzinger et al. used several biochemical and structural biology approaches to search for molecules that bind to and inhibit pyridoxal phosphatase.

The experiments showed that a molecule called 7,8-dihydroxyflavone – which was previously found to improve memory and learning in laboratory animals with brain disorders – binds to pyridoxal phosphatase and inhibits its activity. This led to increased bioactive vitamin B6 levels in mouse brain cells involved in memory and learning.

The findings of Brenner et al. suggest that inhibiting pyridoxal phosphatase to increase vitamin B6 levels in the brain could be used together with supplements. The identification of 7,8-dihydroxyflavone as a promising candidate drug is a first step in the discovery of more efficient pyridoxal phosphatase inhibitors. These will be useful experimental tools to directly study whether increasing the levels of bioactive vitamin B6 in the brain may help those with mental health conditions associated with impaired memory, learning and mood.

## Introduction

Vitamin B6 is an essential micronutrient that plays an important role in the nervous system (*Bowling, 2011*; *Wilson et al., 2019*), with the vitamin B6 status affecting cognitive function at any age (*di Salvo et al., 2012*; *Mitchell et al., 2014*). Population studies indicate that low vitamin B6 levels are common among older people (*Malouf and Grimley Evans, 2003*), and suggest that vitamin B6 deficiency may influence memory performance and may contribute to age-related cognitive decline (*Hughes et al., 2017*; *Jannusch et al., 2017*; *Xu et al., 2022*; *Elias et al., 2006*). Vitamin B6 deficiency is also associated with other conditions characterized by impaired learning and memory, including neuropsychiatric disorders (*Tomioka et al., 2018*; *Toriumi et al., 2021*; *Arai et al., 2010*), Alzheimer's disease (*Paul, 2021*), and inflammation (*Ueland et al., 2017*; *Danielski et al., 2018*). Nevertheless, the exact molecular mechanisms linking vitamin B6 to these pathologies are often insufficiently understood, and whether vitamin B6 supplementation improves cognition is unclear (*Mitchell et al., 2014*; *Malouf and Grimley Evans, 2003*; *Wang et al., 2022b*; *Behrens et al., 2020*; *Hassel et al., 2019*; *Rutjes et al., 2018*; *Smith and Refsum, 2016*; *Aisen et al., 2008*; *Douaud et al., 2013*).

The term vitamin B6 encompasses the enzymatically interconvertible compounds pyridoxine, pyridoxamine, pyridoxal (collectively referred to as B6 vitamers), and their phosphorylated forms. Among these, only pyridoxal 5'-phosphate (PLP) is co-enzymatically active. In humans, PLP is known to be required for 44 distinct biochemical reactions, including the biosynthesis and/or metabolism of neurotransmitters, amino acids, lipids, and glucose. In addition, B6 vitamers display antioxidant and anti-inflammatory functions (*Percudani and Peracchi, 2003*; *Eliot and Kirsch, 2004*; *Percudani and Peracchi, 2009*; *Parra et al., 2018*).

Cellular PLP availability in the brain depends on numerous factors, including the intestinal absorption of B6 vitamers, extracellular phosphatases, inter-organ transport and intracellular enzymes, and carriers/scavengers involved in PLP formation and homeostasis (*Wilson et al., 2019*). Specifically, intracellular PLP is formed by the pyridoxal kinase (PDXK)-catalyzed phosphorylation of pyridoxal, or the pyridox(am)ine-5'-phosphate oxidase (PNPO)-catalyzed oxidation of pyridox(am)ine 5'-phosphate to PLP. PLP is highly reactive and can undergo condensation reactions with, e.g., primary amino groups or thiol groups in proteins or amino acids. Although the mechanisms of PLP delivery within

the cells are still largely unknown, it is clear that the intracellular availability of PLP for co-enzymatic functions depends on PLP carriers/scavengers and on the hydrolytic activity of pyridoxal 5'-phosphate phosphatase (PDXP) (*Wilson et al., 2019*; *Ciapaite et al., 2023*; *Fux and Sieber, 2020*; *Jang et al., 2003*; *Jeanclos et al., 2019*).

We have previously shown that the genetic knockout of PDXP (PDXP-KO) in mice increases brain PLP levels and improves spatial memory and learning, suggesting that elevated PLP levels can improve cognitive functions in this model (*Jeanclos et al., 2019*). We therefore reasoned that a pharmacological inhibition of PDXP may be leveraged to increase intracellular PLP levels and conducted a high-throughput screening campaign to identify small-molecule PDXP modulators. Here, we report the discovery and the structural and cellular validation of 7,8-dihydroxyflavone (7,8-DHF) as a preferential PDXP inhibitor. 7,8-DHF is a well-studied molecule in brain disorder models characterized by impaired cognition, and widely regarded as a tropomyosin receptor kinase B (TrkB) agonist with brain-derived neurotrophic factor (BDNF)-mimetic activity (*Liu et al., 2016*). However, a direct TrkB agonistic activity of 7,8-DHF has been called into question (*Wang et al., 2022a*; *Boltaev et al., 2017*; *Pankiewicz et al., 2021*; *Todd et al., 2014*; *Chen et al., 2011*). Our serendipitous discovery of 7,8-DHF as a direct PDXP inhibitor provides an alternative mechanistic explanation for 7,8-DHF-mediated effects. More potent, efficacious, and selective PDXP inhibitors may be useful future tools to explore a possible benefit of elevated PLP levels in brain disorders.

## Results

### PDXP activity controls PLP levels in the hippocampus

The hippocampus is important for age-dependent memory consolidation and learning, and impaired memory and learning is associated with PLP deficiency (*di Salvo et al., 2012*). To study a possible contribution of PDXP and/or PDXK to age-related PLP homeostasis in the hippocampus, we performed western blot analyses in young versus older mice. Unexpectedly, we found that both PDXP and PDXK expression levels were markedly higher in hippocampi of middle-aged than of juvenile animals (*Figure 1a*). These data suggest an accelerated hippocampal PLP turnover in older mice, consistent with previous findings in senescent mice (*Fonda et al., 1980*).

An analysis of total hippocampal PLP levels in PDXP-WT and PDXP-KO mice showed an age-dependent profile. PLP levels appeared to peak around 3 months of age (possibly reflecting PLP-dependent neurotransmitter biosynthesis and metabolism during the postnatal developmental period) and descended back to juvenile levels by 12 months of age in both genotypes. Although total hippocampal PLP levels in PDXP-KO mice also decreased with age, they consistently remained above PLP levels in control mice (*Figure 1b*; two-tailed, unpaired t-test of PLP levels in PDXP-WT vs. PDXP-KO hippocampi, all ages combined: p<0.0001).

PLP is protected from hydrolysis by binding to proteins, and PDXP is expected to dephosphorylate only non-protein-bound PLP (*Gohla, 2019*). To test this, we prepared protein-depleted PLP fractions from PDXP-WT and PDXP-KO hippocampal lysates using 3 kDa molecular weight cutoff centrifugal filters. The quantification of PLP in these fractions demonstrated that PDXP loss indeed only increased the pool of protein-depleted PLP, both in young (18–42 days of age) and in older mice (252–352 days of age, corresponding to mature/middle-aged mice), whereas the levels of protein-bound PLP remained unchanged (*Figure 1c*). While the hippocampal levels of non-protein-bound PLP dropped by about 60% over this time span in PDXP-WT mice, they remained elevated in PDXP-KO mice (~2-fold higher in younger, and ~5-fold higher in older PDXP-KO compared to the respective PDXP-WT; see *Figure 1—source data 3* for exact mouse ages). We conclude that hippocampi of older mice are characterized by a specific decrease in the levels of non-protein-bound PLP, and that this age-dependent PLP loss is dependent on PDXP activity. These observations establish that PDXP is a critical determinant of PLP levels in the murine hippocampus and suggest that intracellular PLP deficiency may be alleviated by PDXP inhibition.

### A high-throughput screening campaign identifies 7,8-DHF as a PDXP inhibitor

Pharmacological small-molecule PDXP inhibitors are currently lacking. To identify PDXP inhibitor candidates, we screened the FMP small-molecule repository containing 41,182 compounds for molecules

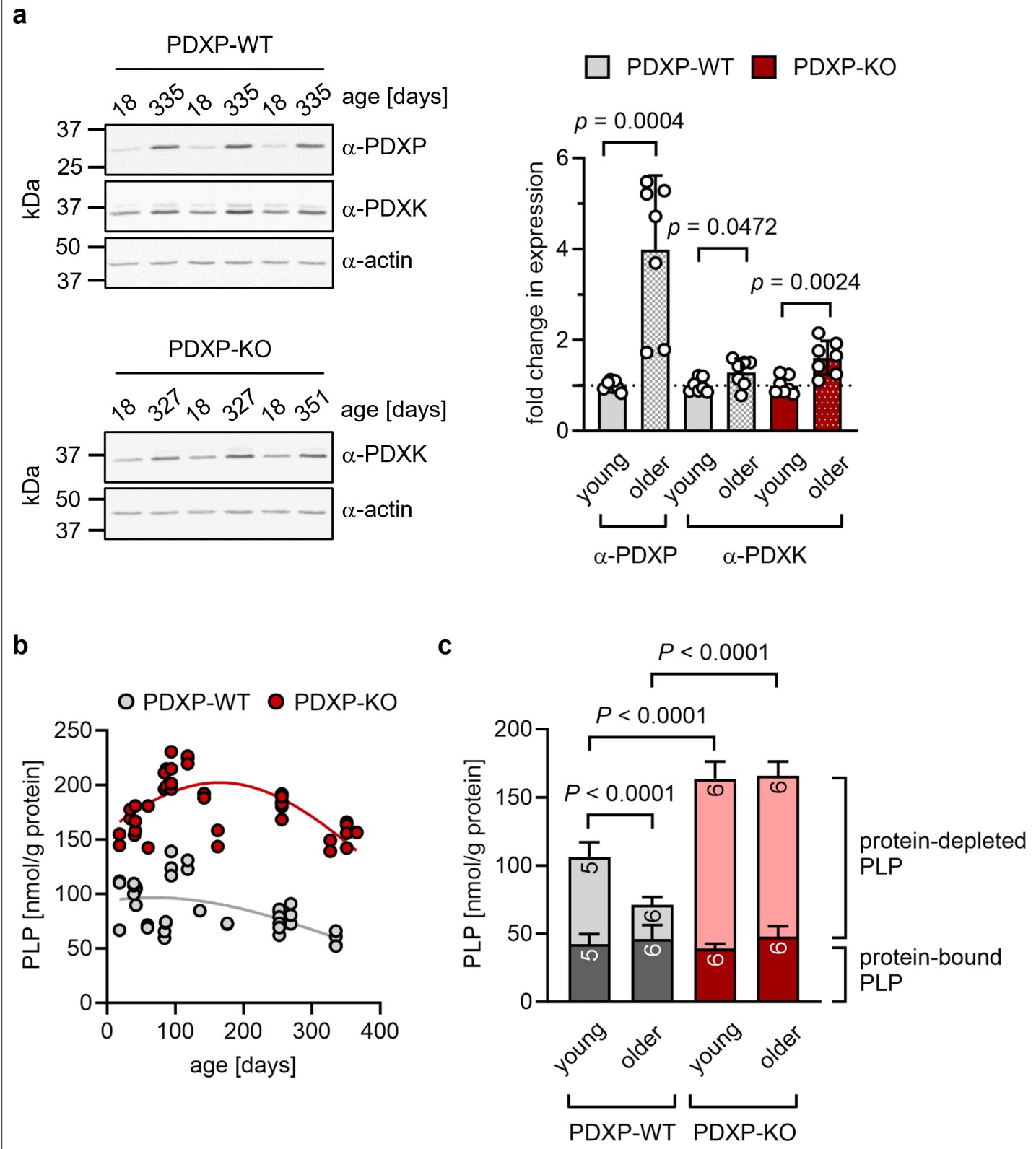

**Figure 1.** Role of pyridoxal 5'-phosphate phosphatase (PDXP) in hippocampal pyridoxal 5'-phosphate (PLP) homeostasis. (**a**) Age-dependent expression of pyridoxal kinase (PDXK) and PDXP in murine hippocampi. *Left panels,* representative western blots of three hippocampi for each genotype. The same blots were reprobed with α-actin antibodies as a loading control. The age of the investigated mice is indicated above the blots. *Right panel,* densitometric quantification of hippocampal PDXP and PDXK western blot signals, corrected by the corresponding actin signals. Young mice were

*Figure 1 continued on next page*

*Figure 1 continued*

18–42 days of age, older mice were 252–351 days of age; n=7 individual hippocampi were analyzed per group. Data are mean values ± SD. Statistical analysis was performed with unpaired, two-sided t-tests; p-values are indicated. (**b**) Age-dependent, total PLP concentrations in isolated hippocampi of PDXP-WT and knockout of PDXP (PDXP-KO) mice. PLP was derivatized with semicarbazide and analyzed by HPLC. Each symbol represents the result of the PLP determination in an individual hippocampus. Data were fitted by Gaussian least-squares analyses. (**c**) Determination of protein-bound and protein-depleted PLP in PDXP-WT and PDXP-KO hippocampal lysates of young (18–42 days of age) and older mice (252–352 days of age). The number of analyzed hippocampi is indicated in the bars. Data are mean values ± SD. Statistical analysis was performed with two-way ANOVA and Tukey's multiple comparisons test. Significant differences (adjusted p-values) in protein-depleted PLP levels are indicated. The exact age of analyzed mice is listed in *Figure 1—source data 3*. Source data are available for this figure.

The online version of this article includes the following source data for figure 1:

**Source data 1.** Western blot quantification (to *Figure 1a*).

**Source data 2.** Quantification of pyridoxal 5'-phosphate (PLP) levels in hippocampi of pyridoxal 5'-phosphate phosphatase (PDXP)-WT and knockout of PDXP (PDXP-KO) mice (to *Figure 1b and c*).

**Source data 3.** Analysis of total hippocampal pyridoxal 5'-phosphate (PLP) levels in pyridoxal 5'-phosphate phosphatase (PDXP)-WT and knockout of PDXP (PDXP-KO) mice.

able to modulate the phosphatase activity of recombinant, highly purified murine PDXP (see *Figure 2—figure supplement 1* for a schematic of the screening campaign). Difluoro-4-methylumbelliferyl phosphate (DiFMUP) was used as a fluorogenic phosphatase substrate in a primary screen. Compounds that altered DiFMUP fluorescence by ≥50% (activator candidates) or ≤25% (inhibitor candidates) were subjected to $EC_{50}/IC_{50}$ value determinations. Of these, 46 inhibitor hits were selected and counter-screened against phosphoglycolate phosphatase (PGP), the closest PDXP relative (*Seifried et al., 2014*; *Jeanclos et al., 2022*). Eleven of the PDXP inhibitor hits (with an $IC_{50}$ PDXP < 20 µM, and $IC_{50}$ PDXP < $IC_{50}$ PGP or no activity against PGP) were subsequently validated in a secondary assay, using PLP as a physiological PDXP substrate (see *Figure 2—source data 5* for all 11 inhibitor hits). Only one PDXP-selective inhibitor hit (7,8-DHF, a naturally occurring flavone) blocked PDXP-catalyzed PLP dephosphorylation ($IC_{50}$ ~ 1 µM).

In vitro activity assays using PLP as a substrate confirmed that 7,8-DHF directly blocks murine and human PDXP activity with submicromolar potency and an apparent efficacy of ~50% (*Figure 2a and b*). We next examined whether commercially available 7,8-DHF analogs might be more potent or efficacious PDXP inhibitors. We tested flavone, 3,7-dihydroxyflavone, 5,7-dihydroxyflavone (also known as chrysin), 3,5,7-trihydroxyflavone (galangin), 5,6,7-trihydroxyflavone (baicalein), and 3,7,8,4'-tetrahydroxyflavone. *Figure 2b* shows that of the tested 7,8-DHF analogs, only 3,7,8,4'-tetrahydroxyflavone was able to inhibit PDXP, albeit with an $IC_{50}$ of 2.5 µM and thus slightly less potently than 7,8-DHF. These results suggest that hydroxyl groups in positions 7 and 8 of the flavone scaffold are required for PDXP inhibition. The efficacy of PDXP inhibition by 3,7,8,4'-tetrahydroxyflavone was not substantially increased at concentrations >40 µM (relative PDXP activity at 40 µM: 0.46±0.05; at 70 µM: 0.38±0.15; at 100 µM: 0.37±0.09; data are mean values ± SD of n=6 experiments). Concentrations >100 µM could not be assessed due to impaired PDXP activity at the DMSO concentrations required for solubilizing the flavone.

We used a biolayer interferometry (BLI) optical biosensing technique to further characterize the binding of 7,8-DHF to PDXP (*Figure 2c*). Consistent with a specific interaction, 7,8-DHF binding to PDXP was concentration-dependent and fully reversible. As a result of the poor solubility of the molecule, a saturation of the binding site was not experimentally accessible. Steady-state analysis of a 7,8-DHF serial dilution series yielded an affinity ($K_D$) value of 3.1±0.3 µM (data are mean values ± SE of n=4 measurements; see *Figure 2—figure supplement 2* for the three other measurements) using a 1:1 dose-response model. Global analysis of the sensorgrams assuming a 1:1 binding model resulted in an affinity of 2.6±0.5 µM, in line with the steady-state results (*Figure 2c*). As expected, 5,7-dihydroxyflavone showed no signal in the BLI, in line with previous experiments (see *Figure 2b*). With its molecular size of 254 Da and its physicochemical properties, 7,8-DHF is a typical fragment-like molecule (*Congreve et al., 2003*). Typical association rate constants ($k_{on}$) for fragments are limited by the rate of diffusion and are higher than $10^6 \cdot M^{-1} s^{-1}$. Interestingly, 7,8-DHF showed a slow $k_{on}$ of $1.05 \cdot 10^4 M^{-1} s^{-1}$, which is atypical and rarely found for fragment-like molecules (*O'Connell et al., 2019*), and a $k_{off}$ rate of 0.03 $s^{-1}$. With the commonly used estimation of ΔG~$pK_D$ and a heavy atom number of 19, 7,8-DHF shows a high ligand efficiency of 0.39, which makes it an interesting molecule

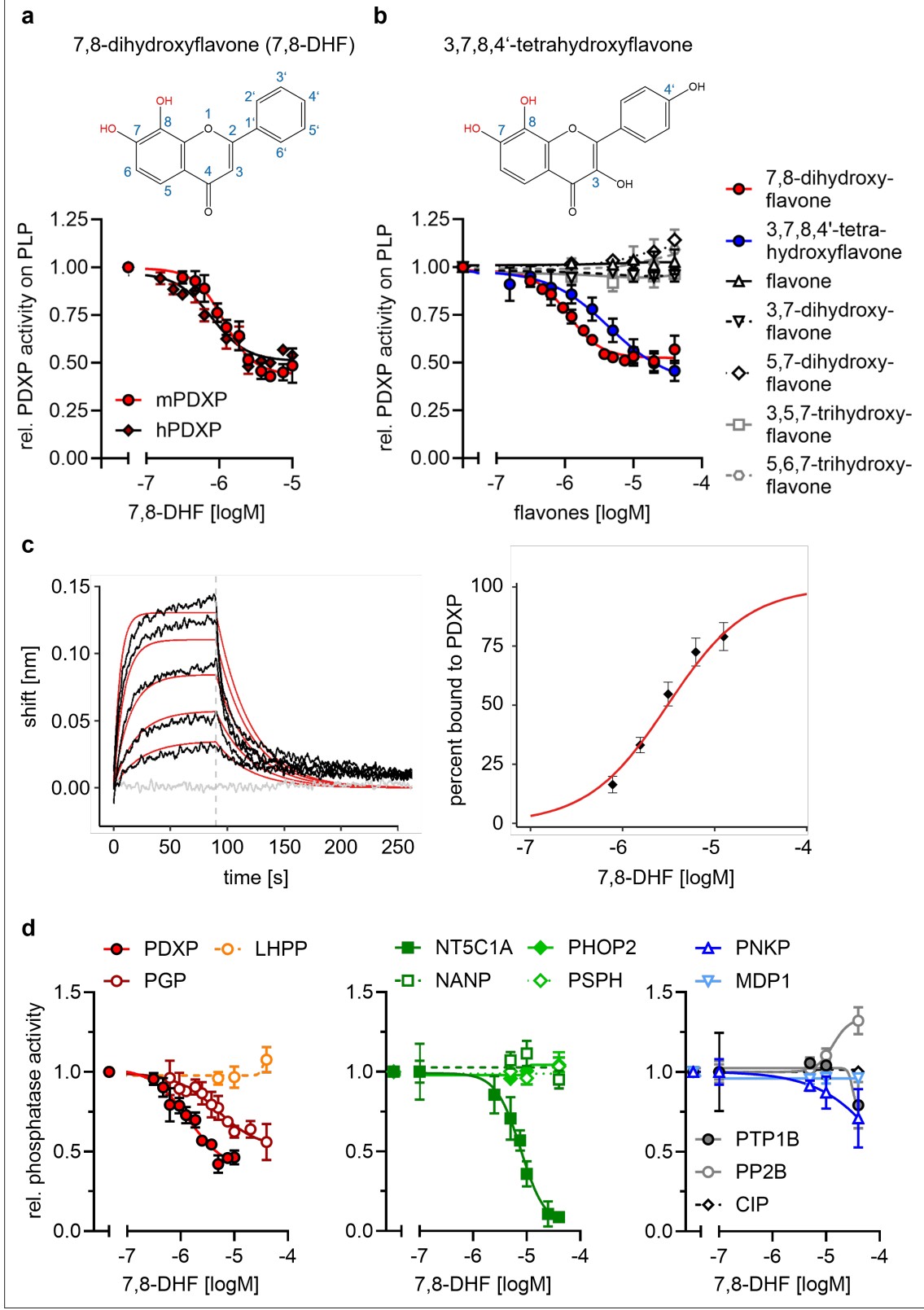

**Figure 2.** Characterization of the 7,8-dihydroxyflavone (7,8-DHF)/pyridoxal 5′-phosphate phosphatase (PDXP) interaction. (**a**) Determination of half-maximal inhibitory constants ($IC_{50}$) of 7,8-DHF (2D structure shown on top) for purified murine or human PDXP, using pyridoxal 5′-phosphate (PLP) as a substrate. Phosphatase activities in the presence of 7,8-DHF were normalized to the respective enzyme activities measured in the presence of the DMSO solvent control. Data are mean values ± SD of n=3 (human PDXP) and n=4 (murine PDXP) biologically independent experiments. (**b**) $IC_{50}$

*Figure 2 continued on next page*

*Figure 2 continued*

values of different flavones for purified murine PDXP with PLP as a substrate. Phosphatase activities in the presence of flavones were normalized to the respective enzyme activities in the presence of the DMSO solvent control. All data are mean values ± SD. The inhibition of PDXP by 3,7,8-trihydroxyflavone-4'-hydroxyphenyl (2D structure shown on top) was assessed in n=6 biologically independent experiments. All other data are from n=3 biologically independent experiments. Apparently missing error bars are hidden by the symbols. (**c**) Biolayer interferometry (BLI) measurements of the interaction of 7,8-DHF with purified murine PDXP. *Left panel*, example sensorgram overlaid with the global 1:1 binding model (red) and the negative control (gray). The dashed line indicates the start of the dissociation phase. *Right panel*, steady-state dose-response analysis for 7,8-DHF based on n=4 technically independent measurements. (**d**) Sensitivity of the indicated phosphatases to 7,8-DHF. Phosphatase activities in the presence of 7,8-DHF were normalized to the respective enzyme activities measured in the presence of the DMSO solvent control. Data are mean values ± SD of n=4 (PGP) or n=3 biologically independent experiments (all other phosphatases). Phosphatase substrates and haloacid dehalogenase (HAD) phosphatase cap types are indicated in parentheses. PDXP, pyridoxal 5'-phosphate phosphatase (pyridoxal 5'-phosphate, C2); PGP, phosphoglycolate phosphatase (2-phosphoglycolate; C2); LHPP, phospholysine phosphohistidine inorganic pyrophosphate phosphatase (imidodiphosphate; C2); NT5C1A, soluble cytosolic 5'-nucleotidase 1A (AMP; C1); NANP, *N*-acetylneuraminate 9-phosphate phosphatase (6-phosphogluconate; C1); PHOP2, phosphatase orphan 2 (pyridoxal 5'-phosphate; C1); PSPH, phosphoserine phosphatase (*O*-phospho-L-serine; C1); PNKP, polynucleotide kinase phosphatase (3-phospho-oligonucleotide; C0); MDP1, magnesium-dependent phosphatase-1 (D-ribose-5-phosphate; C0); PTP1B (protein tyrosine phosphatase 1B; EGFR phospho-peptide); PP2B, protein phosphatase 2B/calcineurin (PKA regulatory subunit type II phospho-peptide); CIP, calf intestinal phosphatase (*p*NPP). Source data are available for this figure.

The online version of this article includes the following source data and figure supplement(s) for figure 2:

**Source data 1.** Phosphatase activity assays (to *Figure 2a*).

**Source data 2.** Phosphatase activity assays (to *Figure 2b*).

**Source data 3.** Biolayer interferometry (BLI) measurements with 7,8-dihydroxyflavone (7,8-DHF) and murine pyridoxal 5'-phosphate phosphatase (PDXP) (to *Figure 2c*).

**Source data 4.** Effect of 7,8-dihydroxyflavone (7,8-DHF) on the phosphatase activity of different phosphatases (to *Figure 2d*).

**Source data 5.** Pyridoxal 5'-phosphate phosphatase (PDXP) inhibitor hits.

**Figure supplement 1.** Identification of pyridoxal 5'-phosphate phosphatase (PDXP) inhibitors.

**Figure supplement 1—source data 1.** Screening campaign for pyridoxal 5'-phosphate phosphatase (PDXP) inhibitors: $IC_{50}$ data of the PDXP_PDXP primary screen.

**Figure supplement 1—source data 2.** Screening campaign for pyridoxal 5'-phosphate phosphatase (PDXP) inhibitors: $IC_{50}$ data of the PGP_PDXP counter-screen.

**Figure supplement 2.** Biolayer interferometry (BLI) measurements of the interaction of 7,8-dihydroxyflavone (7,8-DHF) with purified murine pyridoxal 5'-phosphate phosphatase (PDXP).

for further medicinal chemistry optimization. Taken together, these data support a direct and reversible physical interaction between 7,8-DHF and PDXP that leads to PDXP inhibition.

## Selectivity of 7,8-DHF

PDXP is a member of the large family of haloacid dehalogenase (HAD)-type hydrolases (*Seifried et al., 2013*). HAD phosphatases are $Mg^{2+}$-dependent phospho-aspartate transferases that consist of a Rossman-like catalytic core linked to a cap domain. The insertion site, structure, and size of the cap define the substrate selectivity of the respective enzyme. The 'capless' C0-type HAD phosphatases contain either a very small or no cap, resulting in an accessible catalytic cleft that enables the dephosphorylation of macromolecular substrates. Larger C1 or C2 caps act as a roof for the entrance to the active site; most C1/C2-capped HAD phosphatases consequently dephosphorylate small molecules that can gain access to the catalytic cleft. Cap domains also contain the so-called substrate specificity loops that contribute to substrate coordination. Hence, caps are distinguishing features of HAD phosphatases (*Gohla, 2019*; *Seifried et al., 2013*; *Burroughs et al., 2006*; *Huang et al., 2015*).

To probe the selectivity of 7,8-DHF for PDXP, a C2-capped HAD phosphatase, we tested eight other mammalian HAD phosphatases, including two other C2-, four C1-, and two C0-type enzymes. In addition, we analyzed the activity of 7,8-DHF toward the prototypical tyrosine phosphatase PTP1B (which is known to be sensitive to specific flavonoids, *Proença et al., 2018*); the serine/threonine protein phosphatase calcineurin (PP2B); and a DNA/RNA-directed alkaline phosphatase (calf intestinal phosphatase [CIP]) (*Figure 2d*). When assayed at nominal concentrations of 5, 10, and 40 µM (i.e. up to ~40-fold above the $IC_{50}$ value for PDXP-catalyzed PLP dephosphorylation), 7,8-DHF was completely inactive against six of the tested enzymes. At the highest tested concentration of 40 µM, 7,8-DHF weakly inhibited PTP1B and the polynucleotide kinase-3'-phosphatase PNKP and appeared

**Table 1.** Kinetic constants of pyridoxal 5'-phosphate phosphatase (PDXP)-catalyzed pyridoxal 5'-phosphate (PLP) hydrolysis in the presence of 7,8-dihydroxyflavone (7,8-DHF).

| 7,8-DHF [μM] | 0 | 1.0 | 1.5 | 2.0 | 3.0 | 5.0 | 10.0 |
|---|---|---|---|---|---|---|---|
| $K_M$ [μM] | 14.98 ±1.28 | 18.54 ±6.24 | 20.20 ±6.19 | 18.97 ±5.15 | 24.83 ±2.61 | 32.96 ±2.13 | 30.61 ±2.57 |
| $v_{max}$ [μmol/min/mg] | 1.08 ±0.04 | 0.95 ±0.01 | 0.89 ±0.04 | 0.85 ±0.02 | 0.80 ±0.04 | 0.81 ±0.05 | 0.74 ±0.06 |
| $k_{cat}$ [$s^{-1}$] | 0.57 ±0.02 | 0.5 ±0.01 | 0.47 ±0.02 | 0.45 ±0.01 | 0.42 ±0.02 | 0.43 ±0.03 | 0.39 ±0.03 |
| $k_{cat}/K_M$ [$s^{-1} \cdot M^{-1}$] ($\times 10^{-4}$) | 3.93 ±0.29 | 3.27 ±0.84 | 2.75 ±0.67 | 2.72 ±0.66 | 1.72 ±0.10 | 1.31 ±0.06 | 1.29 ±0.01 |

The data are mean values ± SEM of n=3 technically independent experiments, except for the solvent control samples (n=6). Curves were fitted and parameters $K_M$ (Michaelis-Menten constant); $v_{max}$ (maximum enzyme velocity); $k_{cat}$ (turnover number) were derived using the Michaelis-Menten model in GraphPad Prism 9.5.1. The $k_{cat}$ values were calculated from the maximum enzyme velocities using a molecular mass of 31,828 Da for PDXP. DMSO concentrations were kept constant (0.1% DMSO under all conditions, including the solvent control samples). Source data are available for this table.

The online version of this article includes the following source data for table 1:

**Source data 1.** Kinetic constants of pyridoxal 5'-phosphate phosphatase (PDXP)-catalyzed pyridoxal 5'-phosphate (PLP) hydrolysis in the presence of 7,8-dihydroxyflavone (7,8-DHF).

to increase the activity of calcineurin. As expected, 7,8-DHF inhibited PGP, the closest PDXP relative, with an $IC_{50}$ value of 4.8 μM. This result is consistent with the criteria applied during the initial counter-screen (see above). In addition to PGP, 7,8-DHF inhibited the C1-capped soluble cytosolic 5'-nucleotidase 1A (NT5C1A) with an $IC_{50}$ value of ~10 μM. NT5C1A is an AMP hydrolase expressed in skeletal muscle and heart (*Bianchi and Spychala, 2003*), which is also sensitive to inhibition by small molecules that target the closest PDXP-relative PGP (*Jeanclos et al., 2022*). Together, the selectivity analysis of 7,8-DHF on a total of 12 structurally and functionally diverse protein- and non-protein-directed phosphatases show that 7,8-DHF preferentially inhibits PDXP, and that higher 7,8-DHF concentrations can also target the PDXP paralog PGP and the nucleotidase NT5C1A.

## Mode of PDXP inhibition

To probe the mechanism of PDXP inhibition, we assayed the steady-state kinetics of PLP dephosphorylation in the presence of increasing 7,8-DHF concentrations (*Table 1*). Analysis of the derived kinetic constants demonstrated that 7,8-DHF increased the $K_M$ up to ~2-fold, and slightly reduced $v_{max}$ values ~0.7-fold. Thus, 7,8-DHF mainly exhibits a mixed mode of PDXP inhibition, which is predominantly competitive.

## Co-crystal structures of PDXP bound to 7,8-DHF

To investigate the mechanism of PDXP inhibition in more detail, we co-crystallized homodimeric, full-length murine and human PDXP (mPDXP, hPDXP) with this compound. 7,8-DHF-bound murine PDXP co-crystallized with phosphate in the cubic space group I23, with protomer A containing the inhibitor and protomer B representing an inhibitor-free state (*Figure 3—figure supplement 1a*). The structure was refined following molecular replacement with full-length murine PDXP (here referred to as apo-mPDXP; Protein Data Bank/PDB entry 4BX3) to a resolution of 2.0 Å resulting in an $R_{work}$ of 18.4% and an $R_{free}$ of 21.1% (PDB code 8QFW). We additionally obtained two co-crystal structures of human PDXP with 7,8-DHF: one in a phosphate-containing and one in a phosphate-free form. Both forms crystallized in the tetragonal space group $P4_3 2_1 2$, and each protomer of both structures contained the inhibitor (*Figure 3a*). These structures were refined following molecular replacement with full-length human PDXP (PDB entry 2P27, here referred to as apo-hPDXP) to a resolution of 1.5 Å resulting in an $R_{work}/R_{free}$ of 17.0/19.4% (phosphate-bound 7,8-DHF-hPDXP, PDB code 9EM1), and to a resolution of 1.5 Å resulting in an $R_{work}/R_{free}$ of 18.2/20.5% (phosphate-free 7,8-DHF-hPDXP, PDB code 8S8A). Data collection and refinement statistics are summarized in *Table 2*.

Like their respective apo-forms, 7,8-DHF-bound murine and human PDXP homodimerize via their cap domains (*Figure 3a* and *Figure 3—figure supplement 1a*). The Cα atom-based alignment of the

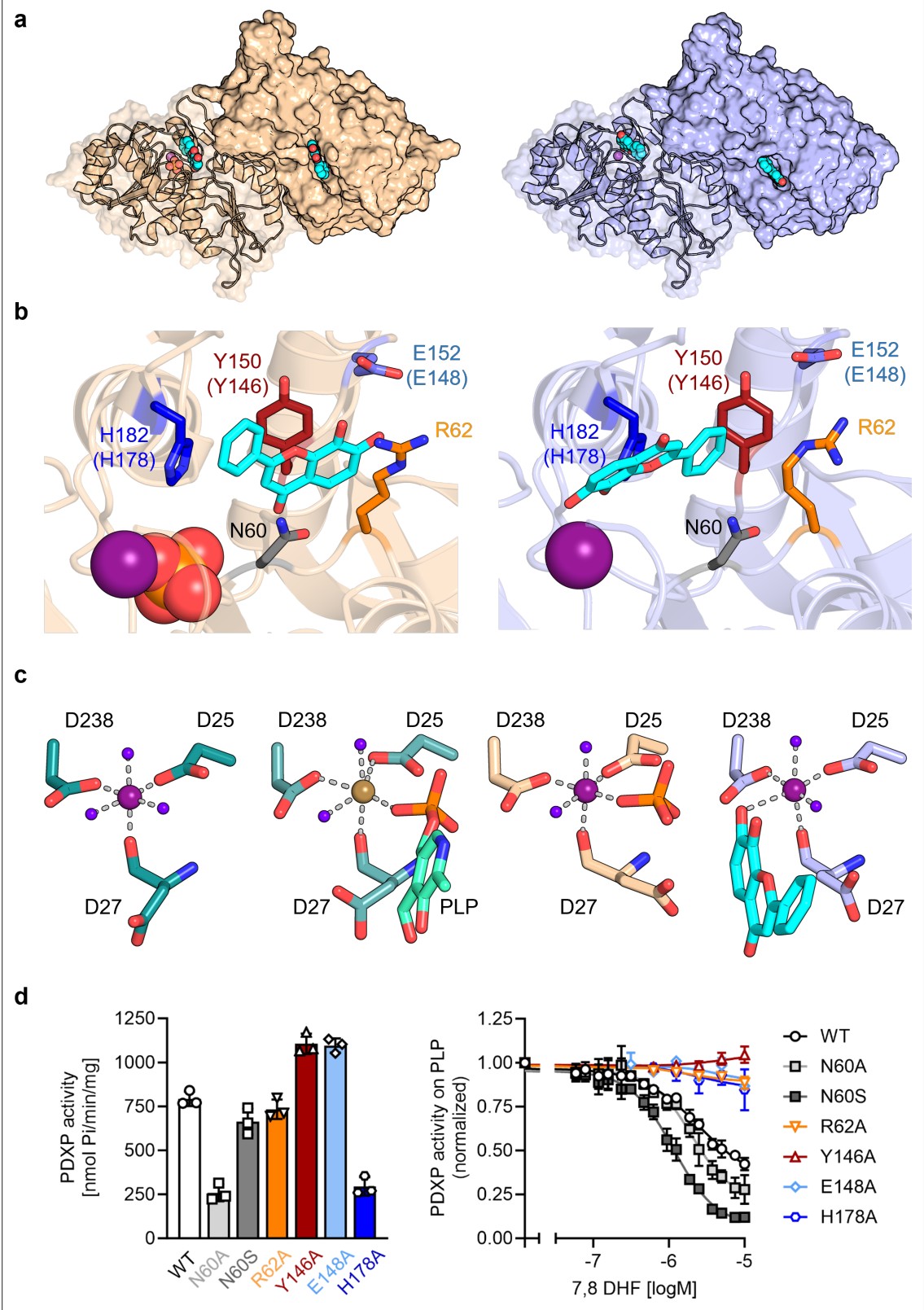

**Figure 3.** X-ray crystal structures of human pyridoxal 5'-phosphate phosphatase (PDXP) in complex with 7,8-dihydroxyflavone (7,8-DHF). (**a**) The models were refined to a resolution of 1.5 Å for full-length human 7,8-DHF-PDXP with phosphate (PDB code 9EM1, colored in wheat yellow, *left panel*) and 1.5 Å for full-length human 7,8-DHF-PDXP without phosphate (PDB code 8S8A, colored in light blue, *right panel*). One protomer of each homodimeric PDXP is shown in cartoon representation and the other protomer in surface representation. 7,8-DHF is displayed in sphere representation with its C-atoms

*Figure 3 continued*

in cyan. Mg$^{2+}$ ions are shown as deep purple spheres and phosphate ions are shown in sphere representation with the phosphorous atom in orange. (**b**) Orientation of 7,8-DHF in the active sites of human 7,8-DHF-PDXP in the presence or absence of phosphate. Structural details of bound 7,8-DHF and adjacent residues of the active sites are shown. *Left,* phosphate-containing 7,8-DHF-PDXP (wheat yellow, cartoon representation). *Right,* phosphate-free 7,8-DHF-PDXP (light blue, cartoon representation). 7,8-DHF is shown in stick representation (cyan C-atoms). The corresponding amino acids in murine PDXP are given in parentheses (see also *Figure 3—figure supplement 1e and f*). (**c**) Comparison of the Mg$^{2+}$ coordination spheres. *From left to right*: human apo-PDXP (PDB: 2P27), human PDXP in complex with pyridoxal 5'-phosphate (PLP) (PDB: 2CFT), human PDXP in complex with 7,8-DHF in the presence of phosphate (PDB: 9EM1), human PDXP in complex with 7,8-DHF in the absence of phosphate (PDB: 8S8A). The catalytically essential Mg$^{2+}$ is shown as a deep purple sphere. In 2CFT, Mg$^{2+}$ was exchanged for Ca$^{2+}$, which is shown here as a light brown-colored sphere. Water molecules are shown as blue spheres. (**d**) Verification of 7,8-DHF-PDXP interactions. *Left panel*, phosphatase activity of purified PDXP or the indicated PDXP variants. Data are mean values ± SD of n=3 biologically independent experiments. *Right panel*, determination of the IC$_{50}$ values of 7,8-DHF for purified PDXP or the indicated PDXP variants. Data are mean values ± SD of n=3 biologically independent experiments. Apparently missing error bars are hidden by the symbols.

The online version of this article includes the following source data and figure supplement(s) for figure 3:

**Source data 1.** Phosphatase activity and 7,8-dihydroxyflavone (7,8-DHF) sensitivity of pyridoxal 5'-phosphate phosphatase (PDXP) and PDXP variants (to *Figure 3g*).

**Figure supplement 1.** X-ray crystal structures of murine pyridoxal 5'-phosphate phosphatase (PDXP) in complex with 7,8-dihydroxyflavone (7,8-DHF).

**Figure supplement 1—source data 1.** Effect of 7,8-dihydroxyflavone (7,8-DHF) on the phosphatase activity of pyridoxal 5'-phosphate phosphatase (PDXP)-D14A.

**Figure supplement 2.** 7,8-Dihydroxyflavone (7,8-DHF) coordination in pyridoxal 5'-phosphate phosphatase (PDXP).

**Figure supplement 3.** Alignment of human and murine pyridoxal 5'-phosphate phosphatase (PDXP).

**Figure supplement 4.** Salt bridge formation between Glu152 (Glu148) and Arg62 gates the active site entrance in pyridoxal 5'-phosphate phosphatase (PDXP).

**Figure supplement 5.** Purity of the employed pyridoxal 5'-phosphate phosphatase (PDXP) and PDXP variants.

structures representing murine apo-PDXP and murine 7,8-DHF-bound PDXP resulted in root mean square (RMS) deviations in the range of 0.43–0.71 Å. Even smaller values were obtained when human apo-PDXP, human PLP-bound PDXP, and human 7,8-DHF-bound PDXP were superimposed with RMS deviations in the range from 0.29 to 0.54 Å (*Table 3*). Hence, binding of the inhibitor did not result in significant changes in murine or human PDXP backbone conformations. All catalytic core residues and the Mg$^{2+}$ cofactor are correctly oriented in the presence of the inhibitor. We conclude that 7,8-DHF binding does not appear to impact the overall fold of murine or human PDXP.

7,8-DHF was observed to only bind to one subunit (the A-chain) of murine PDXP (*Figure 3—figure supplement 1a*) with well-defined density (*Figure 3—figure supplement 1b*) and full occupancy since its average B-factor of 45.8 Å$^2$ closely matches the B-factors of the surrounding atoms. Binding to the other subunit (B-protomer) is prevented by a salt bridge between Arg62 and Asp14 of a symmetry-related A-protomer in the crystal (*Figure 3—figure supplement 1c*). The $\chi_1$ and $\chi_2$ torsion angles of the Arg62 side chain observed in the B-protomer correspond to those observed for this side chain in both protomers of the murine apo-structure (4BX3). To allow binding of the inhibitor, the side chain of Arg62 needs to adopt a completely extended conformation, which is prevented by the salt bridge. However, preventing mPDXP salt bridge formation by mutating Asp14 to Ala did not alter the efficacy of 7,8-DHF inhibition (*Figure 3—figure supplement 1d*; see also *Figure 3d* for the characterization of the PDXP-Arg62Ala variant). It is therefore currently unclear whether the mPDXP crystal state with only a single inhibitor bound per dimer reflects the state in solution. Due to the limited solubility of 7,8-DHF, we were unable to address the stoichiometry of 7,8-DHF binding to the PDXP dimer with isothermal calorimetry. It is conceivable that the mPDXP crystal packing is very stable (indeed, 7,8-DHF-bound mPDXP crystallized in the same cubic space group as apo-mPDXP, see *Kestler et al., 2014*, including the aforementioned salt bridge between Arg62 of the B-subunit and Asp14 of a symmetry-related molecule), and that the free energy generated by the formation of the crystal lattice is higher than the free energy generated upon inhibitor binding.

In contrast to murine PDXP, 7,8-DHF bound to human PDXP with a ratio of two inhibitors per homodimer (*Figure 3a*) and well-defined density (*Figure 3—figure supplement 2a*). Interestingly, the orientation of the inhibitor was markedly affected by the presence or absence of phosphate (*Figure 3b*). In the presence of phosphate, the inhibitor moiety that is closest to the Mg$^{2+}$ cofactor is the uncharged phenyl ring of 7,8-DHF. In the absence of phosphate, the inhibitor is flipped horizontally, with the

**Table 2.** Data collection and refinement statistics.

| | mPDXP-7,8-DHF with phosphate (8QFW) | hPDXP-7,8-DHF with phosphate (9EM1) | hPDXP-7,8-DHF without phosphate (8S8A) |
|---|---|---|---|
| **Data collection** | | | |
| Space group | I23 | $P4_32_12$ | $P4_32_12$ |
| a, b, c (Å) | 167.01, 167.01, 167.01 | 53.96, 53.96, 211.75 | 54.04, 54.04, 212.49 |
| α, β, γ (°) | 90, 90, 90 | 90, 90, 90 | 90, 90, 90 |
| Resolution (Å) | 47.21–2.00 (2.07–2.00) | 48.08–1.50 (1.53–1.50) | 48.17–1.50 (1.53–1.50) |
| $R_{sym}$* | 0.190 (4.101) | 0.126 (4.893) | 0.081 (3.670) |
| $R_{pim}$† | 0.030 (0.652) | 0.018 (0.689) | 0.017 (0.731) |
| $CC_{1/2}$ | 1.00 (0.459) | 0.991 (0.526) | 0.999 (0.579) |
| $<I/\sigma I>$‡ | 20.7 (1.1) | 24.6 (1.4) | 17.5 (1.1) |
| Completeness | 0.998 (0.973) | 1.00 (1.00) | 1.00 (1.00) |
| Redundancy | 41.0 (38.5) | 50.7 (51.0) | 25.6 (26.0) |
| **Refinement** | | | |
| Resolution (Å) | 20.00–2.00 (2.07–2.00) | 38.16–1.50 (1.55–1.50) | 48.165–1.50 (1.55–1.50) |
| R-work§ | 0.1838 (0.300) | 0.1702 (0.2740) | 0.1817 (0.2938) |
| R-free¶ | 0.21 (0.322) | 0.1939 (0.3217) | 0.2050 (0.2955) |
| **RMS deviations in** | | | |
| Bond lengths (Å) | 0.002 | 0.012 | 0.005 |
| Bond angles (°) | 0.50 | 1.08 | 0.80 |
| Chiral centers (Å³) | 0.038 | 0.065 | 0.046 |
| Planar groups (Å) | 0.005 | 0.014 | 0.010 |
| Estimated coordinate error (Å) | 0.26 | 0.16 | 0.19 |
| Ramachandran statistics (%) | 98.95/1.05/0 | 98.63/1.37/0 | 98.63/1.37/0 |

Numbers in parentheses refer to the highest resolution data shell.

Ramachandran statistics reflect the percentage of residues in favored/allowed/outlier regions. Source data (raw diffraction images) have been deposited in the Xtal Raw Data Archive and can be accessed under the XRDA entries 8QFW (https://xrda.pdbj.org/entry/8qfw), 9EM1 (https://xrda.pdbj.org/entry/9em1), and 8S8A (https://xrda.pdbj.org/entry/8s8a).

*$R_{sym} = \Sigma_{hkl} \Sigma_i | I_i – <I> |/ \Sigma_{hkl} \Sigma_i I_i$ where $I_i$ is the ith measurement and $<I>$ is the weighted mean of all measurements of I.

†$R_{pim} = \Sigma_{hkl} 1/(N – 1)^{1/2} \Sigma_i|I_i(hkl) – I(hkl)|/ \Sigma_{hkl} \Sigma_i I(hkl)$, where N is the redundancy of the data and I(hkl) the average intensity.

‡$<I/\sigma I>$ indicates the average of the intensity divided by its standard deviation.

§$R_{work} = \Sigma_{hkl} ||F_o| – |F_c||/ \Sigma_{hkl}|F_o|$ where $F_o$ and $F_c$ are the observed and calculated structure factor amplitudes.

¶$R_{free}$ same as R for 5% of the data randomly omitted from the refinement. The number of reflections includes the $R_{free}$ subset.

hydroxylated chromone substructure of 7,8-DHF now located closest to the $Mg^{2+}$ ion (*Figure 3b*, compare *left* and *right panels*). The inhibitor localization in the presence of phosphate was identical in human and murine PDXP (*Figure 3—figure supplement 2e*). The localization of the phosphate ion that co-crystallized with 7,8-DHF-bound human or murine PDXP overlaps exactly with the localization of the PLP phosphate moiety introduced from PDB code 2CFT (human PDXP in complex with PLP) for visualization purposes (*Figure 3c* and *Figure 3—figure supplement 2b*), indicating that the phosphate ion is bound in a catalytically relevant position.

Irrespective of the orientation of 7,8-DHF in the PDXP active site, the inhibitor is embedded in a cavity that is exclusively formed by the active site of protomer A, without a contribution of the dimerization interface with protomer B. All PDXP residues found to engage in 7,8-DHF interactions are identical in murine and human PDXP (*Figure 3—figure supplement 3*). One side of this cavity is formed by the more polar residues Asp27, Asn60, Ser61, and Arg62 (identical amino acid residue numbering in mPDXP and hPDXP), whereas the opposite side is established by the more hydrophobic

**Table 3.** Alignment of murine and human pyridoxal 5'-phosphate phosphatase (PDXP) structures.

|  | 7,8-DHF-mPDXP (+P), protomer A | 7,8-DHF-mPDXP (– P), protomer B |
| --- | --- | --- |
| 7,8-DHF-mPDXP (+P), protomer B | 0.43 Å | |
| Apo-mPDXP, protomer A | 0.50 Å | 0.71 Å |
| Apo-mPDXP, protomer B | 0.60 Å | 0.69 Å |
|  |  |  |
|  | 7,8-DHF-hPDXP (+P) | 7,8-DHF-hPDXP (–P) |
| 7,8-DHF-hPDXP (–P) | 0.37 Å | |
| Apo-hPDXP | 0.54 Å | 0.45 Å |
| PLP-hPDXP | 0.39 Å | 0.29 Å |

Cα atom-based alignment of the structures representing murine apo-PDXP (PDB: 4BX3), 7,8-dihydroxyflavone (7,8-DHF)-bound murine PDXP (with inhibitor-bound protomer A and inhibitor-free protomer B; PDB: 8QFW), human apo-PDXP (PDB: 2P27), 7,8-DHF-bound human PDXP with phosphate (+P) (PDB: 9EM1), 7,8-DHF-bound human PDXP without phosphate (–P) (PDB: 8S8A) and pyridoxal 5'-phosphate (PLP)-bound human PDXP (PDB: 2CFT); mPDXP, murine PDXP; hPDXP, human PDXP. Root mean square deviations are indicated.

residues Tyr150, His182, Pro183, and Leu184 (corresponding to Tyr146, His178, Pro179, and Leu180 in mPDXP). Adjacent to this hydrophobic stretch, the polar residue Glu152 (Glu148 in mPDXP) is located at the active site entrance, directly opposite of Arg62 on the more polar side of the 7,8-DHF binding channel (*Figure 3b* and *Figure 3—figure supplement 1e*).

Interestingly, Glu152 (or Glu148) and Arg62 can form an intramolecular salt bridge that obstructs the active site entrance (*Figure 3—figure supplement 4*). This interaction was observed in phosphate-free 7,8-DHF-hPDXP and phosphate-free PLP-hPDXP, as well as in apo-hPDXP and apo-mPDXP. In contrast, the 7,8-DHF binding pose that is dictated by the concomitant binding of phosphate and 7,8-DHF interferes with the Glu152 (Glu148)-Arg62 interaction in both, hPDXP and mPDXP (*Figure 3—figure supplement 4*). Thus, although we did not find evidence for major cap/core or substrate specificity loop movements (*Jeanclos et al., 2022*; *Kestler et al., 2014*) in PDXP, the presence or absence of a salt bridge formed between the cap domain residue Glu152 (Glu148) and the core domain residue Arg62 indicates subtle conformational changes in PDXP that may mediate an opening or a closure of the active site entrance.

Inhibitor binding in the presence of phosphate is identical in human and murine PDXP (*Figure 3—figure supplement 1f*) and appears to be primarily stabilized by two hydrogen bonds, as well as polar and non-polar interactions (*Figure 3b*, *left panel*). The side chain hydroxyl group of Ser61 forms a direct hydrogen bond with the ketone group of the inhibitor, which is additionally coordinated by the Ser61 backbone nitrogen atom. Furthermore, Glu152 (Glu148) forms a direct hydrogen bond via its carboxylic acid with the 7-hydroxyl group of 7,8-DHF. The side chains of the polar residues Asp27, Asn60, and Arg62 engage in van der Waals interactions with 7,8-DHF. The two hydroxyl groups of the 7,8-DHF benzyl ring engage in van der Waals interactions with the guanidinium group of Arg62 and the carboxylic acid function of Glu148 (Glu152). On the more hydrophobic side of the binding cavity, Tyr146 (Tyr150) forms π-electron stacking interactions with the pyrone ring of 7,8-DHF. In addition, the His178 (His182) imidazole group coordinates the 7,8-DHF phenyl ring via a cation-π interaction. His178 (His182), located in the substrate specificity loop, and Asn60 and Arg62 are also important for PLP binding (*Kestler et al., 2014*; *Knobloch et al., 2015*).

Inhibitor binding in the absence of phosphate is primarily stabilized by metal coordination and hydrogen bonds, as well as polar and non-polar interactions (*Figure 3b*, *right panel*). The 7-hydroxyl group of 7,8-DHF is involved in an octahedral Mg$^{2+}$ coordination, albeit with an elongated oxygen-Mg$^{2+}$ distance of 2.7 Å, leading to the displacement of a water molecule (*Figure 3c*). This inhibitor-based water displacement is not observed in the phosphate-containing murine or human 7,8-DHF-PDXP structures. The 8-hydroxyl group of 7,8-DHF forms a hydrogen bond with Arg239 and His182. The ketone group of the inhibitor participates in a water-bridged hydrogen bond to the carboxyl group of Asp27 and the backbone amine group of Gly33. Like in the phosphate-containing structure, the His182 imidazole group coordinates the 7,8-DHF phenyl ring via π-π stacking. In addition, Tyr150 forms edge to face π-π stacking interactions with the phenyl ring of 7,8-DHF. The side chains of the

polar residues Asp27, Asn60, and, to some degree, also of Arg62, engage in van der Waals interactions with 7,8-DHF.

To verify the putative 7,8-DHF-PDXP interactions, we introduced single mutations into the binding interface. Asn60, Arg62, Tyr146, Glu148, and His178 in mPDXP were each exchanged for Ala (PDXP$^{N60A}$, PDXP$^{R62A}$, PDXP$^{Y146A}$, PDXP$^{E148A}$, or PDXP$^{H178A}$, respectively). Since the carboxamide group of Asn60 can form a hydrogen bond with the carboxylate moiety of Asp27, and a loss of this interaction in the PDXP$^{N60A}$ variant is predicted to alter the PDXP structure, we additionally mutated Asn60 to Ser (PDXP$^{N60S}$). PDXP variants were recombinantly expressed and purified from *Eschericha coli* (see *Figure 3—figure supplement 5* for protein purity). *Figure 3d* (left panel) shows that all PDXP variants were enzymatically active. As expected, the phosphatase activities of PDXP$^{N60A}$ and of PDXP$^{H178A}$ were reduced. The somewhat elevated phosphatase activity of PDXP$^{Y146A}$ and PDXP$^{E148A}$ is currently unexplained. Importantly, all variants except PDXP$^{N60A}$ and PDXP$^{N60S}$ were resistant to 7,8-DHF, supporting the essential role of each of these residues for inhibitor binding and the minor contribution of the weak van der Waals interactions between Asn60 and 7,8-DHF during inhibitor binding (*Figure 3d*, right panel). These data also suggest that Asn61 contributes to the limited efficacy of 7,8-mediated PDXP inhibition in vitro.

Based on the inhibitor-bound structures and the predominantly competitive component of PDXP inhibition by 7,8-DHF (increased $K_M$, see *Table 1*), it seems likely that 7,8-DHF sterically hinders substrate access to the active site, and competes with PLP coordination (*Figure 3—figure supplement 2b*). In addition, BLI measurements (see *Figure 2c*) showed a relatively slow association rate and extended residence time of 7,8-DHF ($\tau$ =30.3 s). This may indicate a reorganization of the Mg$^{2+}$ coordination due to inhibitor binding, and a reorientation of 7,8-DHF during the PDXP catalytic cycle. The reduced rate of product formation may account for the apparent mixed mode of 7,8-DHF-mediated PDXP inhibition (reduction of $v_{max}$, see *Table 1*).

## 7,8-DHF functions as a PDXP inhibitor in hippocampal neurons

To investigate cellular target engagement of 7,8-DHF, we isolated primary hippocampal neurons from PDXP-WT and PDXP-KO embryos. PDXP deficiency increased total PLP levels 2.4-fold compared to PDXP-WT neurons (*Figure 4a*). This finding is in good agreement with the PLP increase resulting from PDXP loss in total hippocampal extracts (see *Figure 1*). The larger absolute PLP values in cultured neurons are likely attributable to the high concentration of the PLP precursor pyridoxal (20 µM) in the culture medium. We did not observe PDXP-dependent changes in PDXK expression (*Figure 4b*) and could not detect PNPO in hippocampal neuronal cultures, suggesting that the PLP increase was primarily caused by the constitutive PDXP loss.

To assess the consequences of 7,8-DHF treatment on PLP levels in hippocampal neurons, we chose short-term incubation conditions (45 min, 20 µM) to avoid possible secondary effects of the inhibitor. As expected, the acute effect of 7,8-DHF treatment in WT cells was much more subtle (~9% increase in total PLP) than the impact of long-term PDXP deficiency (441.3±62.6 nmol PLP/g protein in DMSO solvent control-treated cells versus 482.7±130.4 nmol PLP/g protein in 7,8-DHF-treated cells; data are mean values ± SE of n=4 independent experiments). However, this effect is likely underestimated because only the PDXP-accessible pool of non-protein-bound PLP may be impacted by 7,8-DHF (see *Figure 1c*). Due to the limited number of available hippocampal neurons, we were unfortunately unable to obtain sufficient quantities of protein-depleted PLP pools to address this question.

Acute changes in the PLP/PL ratio may be a more sensitive indicator of PDXP activity than changes in total PLP levels alone, because PDXP inhibition is expected to increase cellular levels of PLP (the PDXP substrate) and to concomitantly decrease the levels of PL (the product of PDXP phosphatase activity). The PLP/PL ratio is also independent of the exact protein concentration in a given extract of hippocampal neurons, thus optimizing comparability between samples. As shown in *Figure 4*, 7,8-DHF significantly increased the PLP/PL ratio in PDXP-WT, but not in PDXP-KO hippocampal neurons (+18% versus +1% compared to the respective DMSO controls). Together, these data indicate that 7,8-DHF can modulate cellular PLP levels in a PDXP-dependent manner and validate PDXP as a 7,8-DHF target in primary hippocampal neurons.

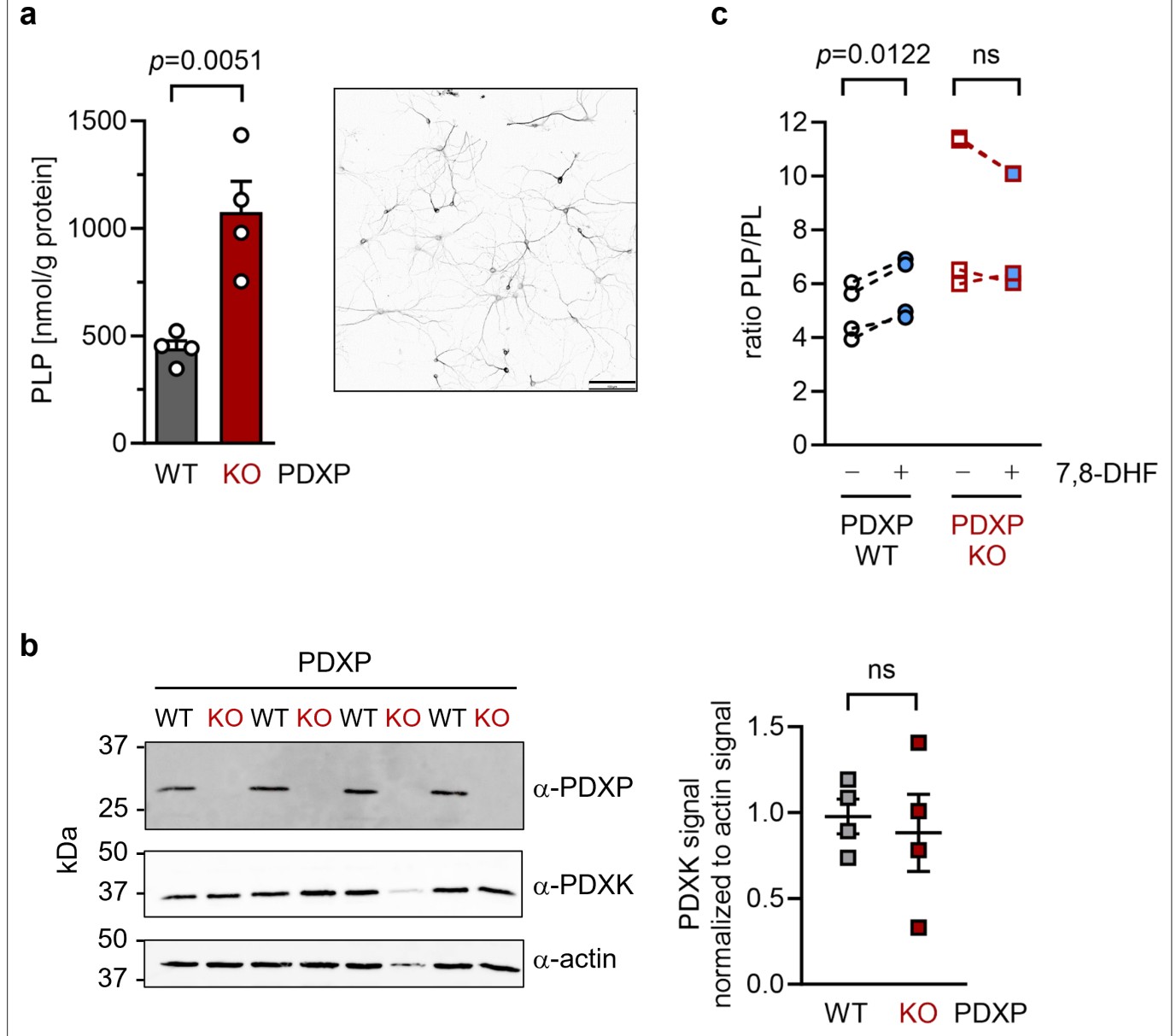

**Figure 4.** Effect of 7,8-dihydroxyflavone (7,8-DHF) on the pyridoxal 5'-phosphate (PLP)/PL ratio in cultured hippocampal neurons from WT or knockout of pyridoxal 5'-phosphate phosphatase (PDXP-KO) mice. (**a**) Effect of long-term PDXP deficiency on total PLP levels in hippocampal neurons. Data are mean values ± SE of n=4 biologically independent experiments. Statistical significance was assessed with a two-tailed, unpaired t-test. A representative image of primary hippocampal neurons stained for the neuronal marker protein MAP2 is shown in the insert (pixel intensities were color-inverted for better visualization). Scale bar, 100 µm. (**b**) Western blot analysis of PDXP and pyridoxal kinase (PDXK) expression in hippocampal neuron samples shown in (**a**). The same blots were reprobed with α-actin antibodies as a loading control. The densitometric quantification of PDXK signals is shown on the right; data are mean values ± SE of n=4 biologically independent experiments. (**c**) Effect of 7,8-DHF (20 µM, 45 min) or the DMSO solvent control (0.02% vol/vol, 45 min) on the PLP/PL ratio in hippocampal neurons of PDXP-WT or PDXP-KO mice. Source data are available for this figure.

The online version of this article includes the following source data for figure 4:

**Source data 1.** Quantification of pyridoxal 5'-phosphate (PLP) and PLP/PL levels in hippocampal neurons (to *Figure 4a and c*).

**Source data 2.** Quantification of western blots (to *Figure 4b*).

## Discussion

PLP deficiency has been associated with human brain disorders for decades (*di Salvo et al., 2012*), yet causal links remain unclear. Aside from vitamin B6 administration, pharmacological strategies to elevate intracellular PLP levels are lacking. Here, we identify 7,8-DHF as a direct PDXP inhibitor that

increases PLP levels in hippocampal neurons, validating PDXP as a druggable target to control intracellular PLP levels in the brain. We also present three high-resolution 7,8-DHF/PDXP co-crystal structures that will facilitate the design of more potent, efficacious, and selective PDXP inhibitors in the future. Such molecules might improve the control of intracellular PLP levels and help to elucidate a possible contribution of PLP to the pathophysiology of brain disorders. Our observation that the expression of PDXP is substantially upregulated in hippocampi of middle-aged mice suggests that a therapeutic vitamin B6 supplementation alone may not suffice to elevate intracellular PLP levels under conditions where the PLP-degrading phosphatase is hyperactive.

The discovery of 7,8-DHF as a direct PDXP inhibitor was unexpected. Interestingly, numerous in vivo studies have reported the effectiveness of 7,8-DHF in brain disorder models, including rodent models of Alzheimer's disease (*Zhang et al., 2014a*; *Devi and Ohno, 2012*; *Bollen et al., 2013*; *Castello et al., 2014*; *Aytan et al., 2018*; *Gao et al., 2016*; *Akhtar et al., 2021*; *Hsiao et al., 2014*), depression (*Blugeot et al., 2011*; *Zhang et al., 2014b*; *Yao et al., 2016*; *Zhang et al., 2016*; *Li et al., 2022*; *Amin et al., 2020*), schizophrenia (*Jaehne et al., 2021*; *Han et al., 2016*; *Yang et al., 2014*; *Han et al., 2017*; *Ren et al., 2013*), epilepsy (*Becker et al., 2015*; *Guarino et al., 2022*), and autism (*Johnson et al., 2012*; *Kang et al., 2017*; *Lee and Han, 2019*; *Chen et al., 2023*). Although PLP deficiency is thought to contribute to the respective human conditions (*di Salvo et al., 2012*; *Majewski et al., 2016*; *Sorolla et al., 2016*), PLP-dependent processes have not yet been considered in the context of 7,8-DHF-induced effects.

7,8-DHF was initially discovered as a small-molecule TrkB agonist with BDNF-mimetic activity (*Jang et al., 2010*). BDNF, a high-affinity TrkB ligand, is an important neuropeptide for nervous system function and pathology. Consensus is emerging that BDNF plays a key role in the treatment response to neuropsychiatric drugs (*Wang et al., 2022a*). Therapeutics that target BDNF/TrkB signaling are thus of interest as disease-modifying agents in several brain disorders. Since BDNF does not cross the blood-brain barrier, attempts have been made to develop small-molecule BDNF mimetics. Several candidates have been reported, including 7,8-DHF (*Boltaev et al., 2017*; *Jang et al., 2010*). Nevertheless, the on-target selectivity and efficacy of these compounds is actively debated. Using quantitative and direct assays to measure TrkB dimerization and activation, TrkB downstream signaling pathways, TrkB-dependent gene expression and cytoprotection, 7,8-DHF and other reported small-molecule TrkB agonists failed to activate TrkB in cells (*Boltaev et al., 2017*; *Pankiewicz et al., 2021*; *Todd et al., 2014*). An electrophysiological study in acute hippocampal slice preparations demonstrated that 7,8-DHF potentiates hippocampal mossy fiber-CA3 synaptic transmission in a TrkB receptor-independent manner (*Kobayashi and Suzuki, 2018*). Overall, it appears that the mechanism of action of 7,8-DHF is incompletely understood, but 7,8-DHF targets other than TrkB so far have remained elusive. The identification of 7,8-DHF as a PDXP inhibitor reported here indicates that this flavone may modulate vitamin B6-dependent processes and suggests that PDXP could be explored as a pharmacological entry point into brain disorders.

## Materials and methods

**Key resources table**

| Reagent type (species) or resource | Designation | Source or reference | Identifiers | Additional information |
|---|---|---|---|---|
| Gene (*Mus musculus*) | Pdxp | UniProtKB | P60487 | |
| Strain, strain background (*Escherichia coli*) | BL21(DE3) pLysS | Stratagene Europe/ VWR | AGLS200132 | |
| Genetic reagent (*M. musculus*; male) | Pdxp[tm1Goh]; C57Bl/6J | Ozgene Ltd.; *Jeanclos et al., 2019* | | Floxed Pdxp mice |
| Genetic reagent (*M. musculus*; female) | B6.FVB-Tg(EIIa-cre) C5379Lmgd/J | Jackson Labs | RRID:MGI:2174520 | Ubiquitous Cre deleter |

*Continued on next page*

*Continued*

| Reagent type (species) or resource | Designation | Source or reference | Identifiers | Additional information |
|---|---|---|---|---|
| Genetic reagent (*M. musculus*; male and female) | Pdxp$^{tm1Goh}$ × EIIa-cre | *Jeanclos et al., 2019* | | Pdxp-deficient mice |
| Biological sample (*M. musculus*) | Primary hippocampal neurons | This paper | | From embryos of Pdxp-deficient or floxed Pdxp control mice |
| Biological sample (*M. musculus*) | Hippocampi | This paper | | Freshly isolated tissues from Pdxp-deficient or floxed Pdxp control mice |
| Antibody | Anti-MAP2 (mouse monoclonal) | Millipore | Cat# MAB3418, RRID:AB_94856 | IF (1:500) |
| Antibody | Anti-actin (mouse monoclonal) | Sigma-Aldrich | Cat# MAB1501, RRID:AB_2223041 | WB (1:5000) |
| Antibody | Anti-PDXP (rabbit monoclonal) | Cell Signaling Technology | Cat# 4686, RRID:AB_2162520 | WB (1:1000) |
| Antibody | Anti-PDXK (rabbit polyclonal) | Sigma-Aldrich | Cat# AV53615, RRID:AB_1855158 | WB (1:1000) |
| Antibody | Anti-PNPO (rabbit polyclonal) | Thermo Fisher Scientific | Cat# PA5-26400, RRID:AB_2543900 | WB (1:1000) |
| Recombinant DNA reagent | pGEX-4T-1 (plasmid) | This paper | | N-terminally GST-tagged, human PDXP |
| Recombinant DNA reagent | pET-SUMO (plasmid) | This paper | | N-terminally His$_6$-SUMO-tagged human PDXP |
| Recombinant DNA reagent | pET-M11 (plasmid) | EMBL Heidelberg | | N-terminally His$_6$-tagged, human SenP2 |
| Recombinant DNA reagent | pET-M11 (plasmid) | *Jeanclos et al., 2022* | | Murine HAD phosphatases (PDXP, PGP, LHPP, NT5C1A, NANP, PHOP2, PSPH, PNKP, MDP1) |
| Sequence-based reagent | Pdxp_F | This paper | PCR primers | TCGACCATGGCGCG CTGCGAGCGG |
| Sequence-based reagent | Pdxp_R | This paper | PCR primers | AAAAGTGAATTCTCAGTC CTCCAGCCCCTC |
| Sequence-based reagent | Pdxp-D14A_F | This paper | PCR primers | GCCCTGCGCGCCGTGCTG GGCCAGGCGCAG |
| Sequence-based reagent | Pdxp-D14A_R | This paper | PCR primers | GCCCAGCACGGCGCGCAGGGC CGCGCCGCG |
| Sequence-based reagent | Pdxp-N60A_F | This paper | PCR primers | TTCGTGAGCAACGCCAGC CGGCGCGCG |
| Sequence-based reagent | Pdxp-N60A_R | This paper | PCR primers | CGCGCGCCGGCTGGC GTTGCTCACGAA |
| Sequence-based reagent | Pdxp-N60S_F | This paper | PCR primers | TTCGTGAGCAACAGCAGCCGGCGCGCG |
| Sequence-based reagent | Pdxp-N60S_R | This paper | PCR primers | CGCGCGCCGGCTGCTG TTGCTCACGAA |
| Sequence-based reagent | Pdxp-R62A_F | This paper | PCR primers | AGCAACAACAGCGCGC GCGCGCGGCCC |
| Sequence-based reagent | Pdxp-R62A_R | This paper | PCR primers | GGGCCGCGCGCGCGC GCTGTTGTTGCT |
| Sequence-based reagent | Pdxp-Y146A_F | This paper | PCR primers | GTGCTCGTAGGCGCC GACGAGCAGTTT |

*Continued on next page*

*Continued*

| Reagent type (species) or resource | Designation | Source or reference | Identifiers | Additional information |
|---|---|---|---|---|
| Sequence-based reagent | Pdxp-Y146A_R | This paper | PCR primers | AAACTGCTCGTCGG CGCCTACGAGCAC |
| Sequence-based Reagent | Pdxp-E148A_F | This paper | PCR primers | GTAGGCTACGACGCGCA GTTTTCCTTC |
| Sequence-based reagent | Pdxp-E148A_R | This paper | PCR primers | GAAGGAAAACTGCGC GTCGTAGCCTAC |
| Sequence-based reagent | Pdxp-H178A_F | This paper | PCR primers | CGCGACCCTTGGG CCCCGCTCAGCGAC |
| Sequence-based reagent | Pdxp-H178A_R | This paper | PCR primers | GTCGCTGAGCGGGG CCCAAGGGTCGCG |
| Peptide, recombinant protein | Bovine brain calcineurin | Sigma-Aldrich | Cat# C1907 | PP2B |
| Peptide, recombinant protein | Phosphopeptide from PKA regulatory subunit type II | Sigma-Aldrich | Cat# 207008 | DLDVPIPGRFDRRVpSVAAE; PP2B substrate |
| Peptide, recombinant protein | Recombinant human PTP1B | Cayman Chemical | Cat# 10010896 | Amino acids 1–321 |
| Peptide, recombinant protein | EGFR phosphopeptide with $Tyr^{992}$ autophosphorylation site | Santa Cruz Biotechnology | Cat# sc-3126 | DADEpYLIPQQG; PTP1B substrate |
| Peptide, recombinant protein | Calf intestinal alkaline phosphatase | NEB | Cat# M0525S | |
| Commercial assay or kit | EZ-Link NHS-PEG4-Biotin | Thermo Fisher | Cat# 21455 | |
| Chemical compound, drug | Flavone; 3,7-dihydroxyflavone; 5,7-dihydroxyflavone; 3,5,7-trihydroxyflavone; 5,6,7-trihydroxyflavone; 7,8-dihydroxyflavone | Sigma-Aldrich | Cat# F2003; Cat# 419826; Cat# 95082; Cat# 282200; Cat# 465119; Cat# D5446 | |
| Chemical compound, drug | 3,7,8,4'-Tetrahydroxyflavone | Ambinter | Cat# AMB30621919 | |
| Software, algorithm | Prism version 9.5.1 | GraphPad Prism | RRID:SCR_002798 | |
| Software, algorithm | OriginPro 2021b | OriginLab | RRID:SCR_014212 | |
| Other | Super Streptavidin Biosensors | Sartorius | Cat# 18-5057 | For biolayer interferometry experiments |

## Materials

Unless otherwise specified, all reagents were of the highest available purity and purchased from Sigma-Aldrich (Schnelldorf, Germany). 3,7,8,4'-Tetrahydroxyflavone was obtained from Ambinter (Orléans, France), all other flavones were from Sigma-Aldrich.

## PDXP knockout mice

Floxed PDXP mice (*Pdxp^tm1Goh*) were generated on a C57Bl/6J background, and whole-body *Pdxp* knockouts were achieved by breeding with B6.FVB-Tg(EIIa-cre)C5379Lmgd/J (EIIa-Cre) transgenic mice, as described (*Jeanclos et al., 2019*). All experiments were authorized by the local veterinary authority and committee on the ethics of animal experiments (Regierung von Unterfranken). All analyses were carried out in strict accordance with all German and European Union applicable laws and regulations concerning care and use of laboratory animals.

## Preparation of hippocampi and hippocampal neurons and immunocytochemistry

Mice were sacrificed by cervical dislocation, and brains were immediately placed on a pre-cooled metal plate and dissected under a Leica M80 binocular (Leica, Wetzlar, Germany). Hippocampi were weighed and flash-frozen in liquid nitrogen. The entire procedure was performed in <3 min. Hippocampal lysates were prepared by the addition of ice-cold PBS (200 µL PBS/10 mg hippocampal wet weight) and homogenized for 1 min in a TissueLyser II instrument (QIAGEN, Hilden, Germany). One fourth of the obtained volume of each lysate was used for the analysis of total PLP concentrations as described below. To determine protein-depleted PLP (*Ciapaite et al., 2023*), the remaining volume of each lysate was centrifuged at 14,000×*g* for 15 min at 4°C. The supernatant was applied to 3 kDa MWCO filters (Amicon Ultra-0.5 Centrifugal Filter; Merck Millipore, Darmstadt, Germany), and centrifuged at 14,000×*g* for 45 min at 4°C. The flow-through was collected and prepared for HPLC analysis (see below).

Primary hippocampal neuronal cultures were prepared from mouse embryos at embryonic day 17. Hippocampi were incubated with 0.5 mg/mL trypsin, 0.2 mg/mL EDTA, and 10 µg/mL DNase I in PBS for 30 min at 37°C. Trypsinization was stopped by adding 10% fetal calf serum. Cells were dissociated by trituration, counted, and seeded at a density of 150,000 cells per 35 mm dish. Dissociated cells were grown in neurobasal medium supplemented with L-glutamine and B27 supplement (A3582801, Life Technologies, Dreieich, Germany) with an exchange of 50% of the medium after 6 days in culture. After 21 days of differentiation (day in vitro 21 [DIV21]), 7,8-DHF (20 µM) or DMSO (0.02%, vol/vol) was added to the hippocampal neuronal cultures for 45 min. Cells were rinsed once with PBS (37°C), lysed in 150 µL ice-cold $H_2O$, and placed at –80°C for at least 30 min.

For immunocytochemistry, DIV21 primary hippocampal neurons were fixed with 4% (wt/vol) paraformaldehyde in phosphate-buffered saline (PBS) for 15 min at room temperature (RT). After washing twice with PBS, 50 mM $NH_4Cl$ was added for 10 min. Cells were then permeabilized with 0.1% (vol/vol) Triton X-100 and blocked with 5% (vol/vol) goat serum in PBS for 30 min at 22°C. Cells were incubated with mouse monoclonal anti-MAP2 antibodies (1:500 dilution, clone AP20, Millipore, Darmstadt, Germany) for 1 hr in 5% goat serum/PBS at 22°C. Alexa488-labeled secondary goat anti-mouse antibodies (1:500 dilution; Dianova, Hamburg, Germany) were applied for 1 hr. Nuclei were counterstained with 4′,6-diamino-2-phenylindole (DAPI), and slides were mounted with Mowiol. Images were acquired using an inverted IX81 microscope equipped with an Olympus UPLSAPO 60× oil objective (numerical aperture: 1.35) on an Olympus FV1000 confocal laser scanning system, using an FVD10 SPD spectral detector and diode lasers of 405 nm (DAPI) and 495 nm (Alexa488).

## Determination of PLP and PL by HPLC

Samples were derivatized as described (*Talwar et al., 2003*). Briefly, 100 µL of lysate were mixed with 8 µL derivatization agent (containing 250 mg/mL of both semicarbazide and glycine), and incubated on ice for 30 min. Samples were then deproteinized by the addition of perchloric acid (8 µL of a 72% [wt/vol] stock solution), followed by centrifugation at 15,000×*g* for 15 min at 4°C. Supernatants (100 µL) were neutralized with 10 µL NaOH (25% [vol/vol] stock solution), and 2 µM pyridoxic acid (PA) was added as an internal standard. PLP and PL were subjected to the same derivatization protocol to establish a standard curve. Samples were analyzed on a Dionex Ultimate 3000 HPLC (Thermo Fisher Scientific, Dreieich, Germany), using 60 mM $Na_2HPO_4$, 1 mM EDTA, 9.5% (vol/vol) MeOH; pH 6.5 as mobile phase. PL, PLP, and pyridoxic acid (PA) were separated on a 3 µm reverse phase column (BDS-HYPERSIL-C18, Thermo Fisher Scientific). Chromatograms were analyzed using Chromeleon 7 software (Thermo Fisher Scientific).

## Western blotting

Tissue or cell homogenates (prepared as detailed above for HPLC analysis) were extracted with 4× RIPA buffer (final concentration, 50 mM Tris, pH 7.5; 150 mM NaCl, 1% [vol/vol] Triton X-100, 0.5% [vol/vol] sodium deoxycholate, 0.1% [wt/vol] SDS, 1 mM 4-(2-aminoethyl)benzenesulfonyl fluoride [Pefabloc], 5 µg/mL aprotinin, 1 µg/mL leupeptin, 1 µg/mL pepstatin) for 15 min at 4°C under rotation, and lysates were clarified by centrifugation (20,000×*g*, 15 min, 4°C). Protein concentrations in the supernatants were determined using the Micro BCA Protein Assay Kit (Thermo Fisher Scientific). Proteins were separated by SDS-PAGE and transferred to nitrocellulose membranes by semidry blotting. The

following antibodies were used: mouse monoclonal α-actin, clone C4, Sigma Aldrich; rabbit monoclonal α-PDXP, clone C85E3, Cell Signaling Technology, Danvers, MA, USA; rabbit polyclonalα-PDXK (AB1), Sigma-Aldrich; and rabbit polyclonal α-PNPO, Thermo Fisher Scientific, as described in *Jeanclos et al., 2019*. Western blots were densitometrically quantified with NIH ImageJ, version 1.45i.

## Phosphatase plasmids and cloning

N-terminally GST-tagged, human PDXP was in pGEX-4T-1 (Amersham Biosciences, Amersham, UK). N-terminally His$_6$-SUMO-tagged human PDXP was cloned into pET-SUMO (coding for human SUMO; a kind gift of Dr. Pedro Friedmann Angeli, Rudolf-Virchow-Center, University of Würzburg, Germany); His$_6$-SenP2 (EMBL Heidelberg) was in pET-M11. All other phosphatases were of murine origin and were subcloned into pET-M11 (EMBL), as described (*Jeanclos et al., 2022*). Murine *Pdxp* point mutants (generated by nested PCR) were subcloned into the *Nco*I (*Pci*I for *Psph*) and *EcoR*I restriction sites of pET-M11, using Q5 Hot Start High-Fidelity DNA Polymerase (New England Biolabs, Frankfurt/Main, Germany). *Pdxp-D14N* was generated with the Platinum SuperFi II DNA Polymerase Mastermix according to mutagenesis protocol A provided by the manufacturer (Thermo Fisher Scientific) and cloned into pET-M11 as described above. Primers were purchased from Eurofins Genomics (Ebersberg, Germany), and all constructs were verified by sequencing (Microsynth Seqlab, Göttingen, Germany).

## Expression and purification of recombinant proteins

Human His$_6$-SUMO-tagged PDXP was grown in ZYP-5052 autoinduction medium for 7 hr at 37°C, followed by 48 hr at 21°C (*Studier, 2005*). All purification steps of murine PDXP and murine PGP were carried out exactly as described (*Jeanclos et al., 2022*). The purification of the His$_6$-SUMO-tagged human PDXP was carried out exactly as described for murine His$_6$-tagged PDXP (*Jeanclos et al., 2022*), except that human SenP2 protease was used to cleave the His$_6$-SUMO-tag. N-terminally His$_6$-tagged PDXP variants and His$_6$-SenP2 were expressed as described for PDXP-WT (*Jeanclos et al., 2022*). With the exception of PDXP-D14, the His$_6$-tag was not cleaved off. Human GST-PDXP was transformed into *E. coli* BL21(DE3) pLysS (Stratagene Europe/VWR, Darmstadt, Germany). Protein expression was induced with 0.5 mM isopropyl β-d-thiogalactopyranoside for 18 hr at 20°C. All subsequent purification steps were carried out at 4°C. Cells were harvested by centrifugation for 10 min at 8000×*g* and resuspended in lysis buffer (100 mM triethanolamine [TEA], 500 mM NaCl; pH 7.4) supplemented with protease inhibitors (EDTA-free protease inhibitor tablets; Roche, Mannheim, Germany) and 150 U/mL DNase I (Applichem, Renningen, Germany). Cells were lysed using a cell disruptor (Constant Systems, Daventry, UK), and cell debris was removed by centrifugation for 30 min at 30,000×*g*. GST-PDXP was batch-purified on a glutathione sepharose 4B resin (GE Healthcare, Uppsala, Sweden). After extensive washing with 25 column volumes of wash buffer (50 mM TEA, 250 mM NaCl; pH 7.4), GST-PDXP was eluted in wash buffer supplemented with 10 mM reduced glutathione, concentrated, and further purified in buffer A (50 mM TEA, 250 mM NaCl, 5 mM MgCl$_2$; pH 7.4) using a HiLoad 16/60 Superdex 200 pg gel filtration column operated on an ÄKTA liquid chromatography system (GE Healthcare).

## High-throughput screen for PDXP modulators

The screening campaign (chemical library, screening protocol, concentration-dependent assays, data analysis) was conducted exactly as described previously (*Jeanclos et al., 2022*), except that the primary screen was done with PDXP, the counter-screen with PGP, and PDXP inhibitor hits were validated using PLP as a physiological PDXP substrate.

## IC$_{50}$ determinations, enzyme kinetics, and compound selectivity

Conditions for enzymatic assays were as previously published (*Jeanclos et al., 2022*), with the following modifications. Bovine brain calcineurin (PP2B, Sigma-Aldrich #C1907) activity against the PKA regulatory subunit type II (phosphopeptide DLDVPIPGRFDRRVpSVAAE; Sigma-Aldrich #207008) was assayed at 37°C in 100 mM NaCl, 50 mM Tris, 6 mM MgCl$_2$, 0.5 mM CaCl$_2$, 0.5 mM DTT, 0.025% (vol/vol) NP40; pH 7.5. Recombinant human PTP1B (amino acids 1–321, Cayman Chemical, Ann Arbor, MI, USA) activity against the Tyr$^{992}$ autophosphorylation site of EGFR (DADEpYLIPQQG; Santa Cruz Biotechnology, Heidelberg, Germany) was assayed at 30°C in 150 mM NaCl, 50 mM 2-(*N*-morpholino)

ethanesulfonic acid, 1 mM EDTA; pH 7.2. Murine PDXP-D14A was assayed exactly like PDXP-WT in 30 mM TEA, 5 mM MgCl$_2$, 30 mM NaCl; pH 7.5, supplemented with 0.01% (vol/vol) Triton X-100.

Flavone stocks were prepared at 10 mM in 100% DMSO. A constant final DMSO concentration of 0.4% was maintained under all conditions, and solvent control samples contained 0.4% DMSO without compounds. Purified phosphatases were pre-incubated for 10 min at RT with serial dilutions of flavones. Dephosphorylation reactions were started by the addition of the indicated substrate; buffer with substrate and the respective flavone but without the enzyme served as a background control. Prior to compound testing, time courses of inorganic phosphate release from the respective phosphatase substrates were conducted to ensure assay linearity. Inorganic phosphate release was detected with a malachite green solution (Biomol Green; Enzo Life Sciences, Lörrach, Germany); the absorbance at 620 nm ($A_{620}$) was measured on an Envision 2104 multilabel reader (Perkin Elmer, Rodgau, Germany). Released phosphate was determined by converting the values to nmol $P_i$ with a phosphate standard curve. Data were analyzed with GraphPad Prism version 9.5.1 (GraphPad, Boston, MA, USA). For $IC_{50}$ determinations, $\log_{inhibitor}$ versus response was calculated (four parameter). To derive $K_M$ and $k_{cat}$ values, data were fitted by nonlinear regression to the Michaelis-Menten equation.

## Biolayer interferometry

PDXP was biotinylated using the EZ-Link NHS-PEG4-Biotin kit, as recommended by the manufacturer (Thermo Fisher Scientific), and loaded on Super Streptavidin Biosensors (SSA) (Sartorius, Göttingen, Germany) as follows. SSA sensors were equilibrated for 1 hr at RT in BLI assay buffer (250 mM TEA, 5 mM MgCl$_2$, 250 mM NaCl, 0.005% [vol/vol] TWEEN-20; pH 7.5), loaded with 200 µg/mL biotinylated PDXP, blocked with 2 µg/mL biocytin, and washed in BLI assay buffer. Reference SSA sensors were blocked with 2 µg/mL biocytin (*Wartchow et al., 2011*). Six point 1:1 serial dilution series of 7,8-DHF and 5,7-DHF were prepared in DMSO, and BLI assay buffer was added to the wells to obtain a 7,8-DHF starting concentration of 25 µM. The final DMSO concentration was 5% (vol/vol). Buffers for baseline, dissociation, and buffer correction wells were supplemented with the same amount of DMSO for identical buffer conditions. Four measurements were carried out per condition, using one sensor set for two measurements. All measurements were conducted on an Octet K2 device (Sartorius) using 96-well plates. Assay settings were as follows: baseline measurement 45 s, association time 90 s, dissociation time 150 s. The resulting data were processed using the double reference method of the Octet analysis software for removal of drifts and well-to-well artifacts. Kinetic analyses were performed using the Octet analysis software. The steady-state analysis was carried out with OriginPro 2021b (OriginLab, Northampton, MA, USA), using a dose-response model for regression. Due to the poor solubility of 7,8-DHF, the highest concentration of 25 µM was not included in the analysis.

## PDXP crystallization and data collection

For co-crystallization with 7,8-DHF, full-length murine PDXP (10 mg/mL in 50 mM TEA; 250 mM NaCl; 5 mM MgCl$_2$; pH 7.4) was supplemented with a threefold molar excess of the flavone. Prism-shaped crystals of 7,8-DHF-bound murine PDXP were grown at 20°C in 0.1 M phosphate citrate (pH 4.2) and 40% (vol/vol) PEG 300 using the sitting-drop vapor diffusion method. Human PDXP crystals were grown at 20°C in 0.1 M Tris (pH 8.5) and 1 M diammonium hydrogen phosphate, or in 0.1 M HEPES (pH 7.0), 15% (vol/vol) Tacsimat pH 7.0 (Hampton Research, Aliso Viejo, CA, USA) and 2% (wt/vol) PEG 3350 using the sitting-drop vapor diffusion method. Crystals were cryoprotected for flash-cooling in liquid nitrogen by soaking in mother liquor containing 25% (vol/vol) glycerol. Diffraction data of murine PDXP in complex with 7,8-DHF were collected from flash-cooled crystals at a temperature of 100 K on beamline BL 14.1 at the BESSY synchrotron (Helmholtz Zentrum Berlin, Germany). Diffraction data of 7,8-DHF bound to human PDXP were collected on beamline ID23-2 at the ESRF (Grenoble, France) (https://data.esrf.fr/doi/10.15151/ESRF-ES-1409594895). Diffraction data were processed using XDS (*Kabsch, 2010*) and further analyzed with Aimless (*Evans and Murshudov, 2013*) of the CCP4 suite (*Winn et al., 2011*). The structures of 7,8-DHF-PDXP were solved by molecular replacement with the program Phaser (*McCoy et al., 2007*) with the structure of the murine PDXP (PDB entry 4BX3) or human PDXP (PDB entry 2P27) as search models, and refined with Phenix (*Adams et al., 2010*). Model building was carried out in COOT (*Emsley et al., 2010*). Structural illustrations were prepared with PyMOL 2.5.1 (*Schrodinger, 2021*).

## Acknowledgements

We thank Carola Seyffarth and Nicole Bader for excellent technical assistance, Dr. Jochen Kuper for collecting the murine PDXP diffraction data, the staff at beamline BL14.1 of the BESSY synchrotron, and the staff at beamline ID23-2 of the ESFR synchrotron for technical support. A part of this work was initially funded by the DFG Collaborative Research Center SFB688 (TP A11 to AG).

## Additional information

### Competing interests

Antje Gohla: A.G. is a recipient of a research project grant from Boehringer Ingelheim International GmbH. This project funding is independent of and has no overlap with the work described in this manuscript. The other authors declare that no competing interests exist.

### Funding

| Funder | Grant reference number | Author |
|---|---|---|
| Deutsche Forschungsgemeinschaft | SFB688, TPA11 | Antje Gohla |

The funders had no role in study design, data collection and interpretation, or the decision to submit the work for publication.

### Author contributions

Marian Brenner, Formal analysis, Investigation, Visualization, Writing – review and editing, Assisted with preparing the revised manuscript; Christoph Zink, Linda Witzinger, Sebastian Bothe, Formal analysis, Investigation, Visualization; Angelika Keller, Kerstin Hadamek, Investigation; Martin Neuenschwander, Resources, Formal analysis, Investigation; Carmen Villmann, Jens Peter von Kries, Resources; Hermann Schindelin, Resources, Formal analysis, Writing – original draft, Writing – review and editing; Elisabeth Jeanclos, Formal analysis, Supervision, Validation, Investigation, Visualization, Writing – original draft; Antje Gohla, Conceptualization, Formal analysis, Supervision, Funding acquisition, Visualization, Writing – original draft, Project administration, Writing – review and editing

### Author ORCIDs

Marian Brenner http://orcid.org/0009-0001-3781-1056
Carmen Villmann http://orcid.org/0000-0003-1498-6950
Hermann Schindelin http://orcid.org/0000-0002-2067-3187
Antje Gohla https://orcid.org/0000-0002-7442-1487

Reviewer #1 (Public Review): https://doi.org/10.7554/eLife.93094.3.sa1
Reviewer #2 (Public Review): https://doi.org/10.7554/eLife.93094.3.sa2
Reviewer #3 (Public Review): https://doi.org/10.7554/eLife.93094.3.sa3
Author response https://doi.org/10.7554/eLife.93094.3.sa4

## Additional files

### Supplementary files
• MDAR checklist

### Data availability

The previously published PDB entry 4BX3 of murine apo-PDXP [http://doi.org/10.2210/pdb4BX3/pdb], 2P27 of human apo-PDXP [http://doi.org/10.2210/pdb2P27/pdb] and 2CFT of PLP-bound human PDXP [http://doi.org/10.2210/pdb2CFT/pdb] are used in this manuscript. X-ray crystallographic data of 7,8-DHF-bound murine PDXP generated in this study have been deposited in the PDB and can be accessed under the PDB entry 8QFW [http://doi.org/10.2210/pdb8QFW/pdb]. X-ray crystallographic data of 7,8-DHF-bound human PDXP generated in this study can be accessed under

the PDB entries 9EM1 (with phosphate) [http://doi.org/10.2210/pdb9EM1/pdb] and 8S8A (without phosphate) [http://doi.org/10.2210/pdb8S8A/pdb]. The corresponding raw diffraction images have been deposited in the Xtal Raw Data Archive and can be accessed under the XRDA entries 8QFW [https://xrda.pdbj.org/entry/8qfw], 9EM1 [https://xrda.pdbj.org/entry/9em1], and 8S8A [https://xrda.pdbj.org/entry/8s8a]. All other data generated or analyzed during this study are included in the manuscript and source data files. All materials are available from the corresponding authors upon reasonable request and without restrictions.

The following datasets were generated:

| Author(s) | Year | Dataset title | Dataset URL | Database and Identifier |
|---|---|---|---|---|
| Schindelin H, Gohla A | 2023 | Murine pyridoxal phosphatase in complex with 7,8-dihydroxyflavone | http://doi.org/10.2210/pdb8QFW/pdb | Worldwide Protein Data Bank, 10.2210/pdb8QFW/pdb |
| Brenner M, Gohla A, Schindelin H | 2024 | Human pyridoxal phosphatase in complex with 7,8-dihydroxyflavone and phosphate | http://doi.org/10.2210/pdb9EM1/pdb | Worldwide Protein Data Bank, 10.2210/pdb9EM1/pdb |
| Brenner M, Gohla A, Schindelin H | 2024 | Human pyridoxal phosphatase in complex with 7,8-dihydroxyflavone without phosphate | http://doi.org/10.2210/pdb8S8A/pdb | Worldwide Protein Data Bank, 10.2210/pdb8S8A/pdb |
| Schindelin H, Gohla A | 2023 | Murine pyridoxal phosphatase in complex with 7,8-dihydroxyflavone | https://xrda.pdbj.org/entry/8qfw | Xtal Raw Data Archive, 8QFW |
| Brenner M, Gohla A, Schindelin H | 2024 | Human pyridoxal phosphatase in complex with 7,8-dihydroxyflavone and phosphate | https://xrda.pdbj.org/entry/9em1 | Xtal Raw Data Archive, 9EM1 |
| Brenner M, Gohla A, Schindelin H | 2024 | Human pyridoxal phosphatase in complex with 7,8-dihydroxyflavone without phosphate | https://xrda.pdbj.org/entry/8s8a | Xtal Raw Data Archive, 8S8A |

The following previously published datasets were used:

| Author(s) | Year | Dataset title | Dataset URL | Database and Identifier |
|---|---|---|---|---|
| Knobloch G, Gohla A, Schindelin H | 2013 | Crystal Structure of murine Chronophin (Pyridoxal Phosphate Phosphatase) | http://doi.org/10.2210/pdb4BX3/pdb | Worldwide Protein Data Bank, 10.2210/pdb4BX3/pdb |
| Kang BS, Cho HJ, Kim KJ, Kwon OS | 2006 | Crystal structure of human pyridoxal 5'-phosphate phosphatase with its substrate | http://doi.org/10.2210/pdb2CFT/pdb | Worldwide Protein Data Bank, 10.2210/pdb2CFT/pdb |
| Ramagopal UA, Freeman J, Izuka M, Toro R, Sauder JM, Burley SK, Almo SG | 2007 | Crystal Structure of Human Pyridoxal Phosphate Phosphatase with Mg$^{2+}$ at 1.9 A resolution | http://doi.org/10.2210/pdb2P27/pdb | Worldwide Protein Data Bank, 10.2210/pdb2P27/Pdb |

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
