## [Editor Report · eLife assessment]

Following small molecule screens, this study provides **convincing** evidence that 7,8 dihydroxyflavone (DHF) is a competitive inhibitor of pyridoxal phosphatase. These results are **important** since they offer an alternative mechanism for the effects of 7,8 dihdroxyflavone in cognitive improvement in several mouse models. This paper is also significant due to the interest in the phosphatases and neurodegeneration fields.

---

## [Referee Report · Reviewer #1 (Public Review)]

Summary:

This manuscript set out to identify selective inhibitors of the pyridoxal phosphatase (PDXP). Previous studies had demonstrated improvements in cognition upon removal of PDXP, and here the authors reveal that this correlates with an increase in pyridoxal phosphate (PLP; PDXP substrate and an active coenzyme form of vitamin B6) with age. Since several pathologies are associated with decreased vitamin B6, the authors propose that PDXP is an attractive therapeutic target in the prevention/treatment of cognitive decline. Following high throughput and secondary small molecule screens, they identify two selective inhibitors. They follow up on 7, 8 dihydroxyflavone (DHF). Following structure-activity relationship and selectivity studies, the authors then solve a co-crystal structure of 7,8 DHF bound to the active site of PDXP, supporting a competitive mode of PDXP inhibition. Finally, they find that treating hippocampal neurons with 7,8 DHF increases PLP levels in a WT but not PDXP KO context. The authors note that 7,8 DHF has been used in numerous rodent neuropathology models to improve outcomes. 7, 8 DHF activity was previously attributed to activation of the receptor tyrosine kinase TrkB, although this appears to be controversial. The present study raises the possibility that it instead/also acts through modulation of PLP levels via PDXP, and is an important area for future work.

Strengths:

The strengths of the work are in the comprehensive, thorough, and unbiased nature of the analyses revealing the potential for therapeutic intervention in a number of pathologies.

Weaknesses:

Potential weaknesses include the poor solubility of 7,8 DHF that might limit its bioavailability given its relatively low potency (IC50 = 0.8 uM), which was not improved by SAR. The solubility issues of 7,8 DHF have been discussed at length in the authors' response to Reviewer #3. In particular, the solubility of 7,8 DHF has been found to be variable due to the concentration and buffer conditions. The 7,8 DHF compound has an extended residence time and the co-crystal structure could aid the design of more potent molecules and would be of interest to those in the pharmaceutical industry. The images related to crystal structure have been improved with additional structural analysis of PDXP in a complex of 7,8-DHF (see revised Figure 3).

---

## [Referee Report · Reviewer #2 (Public Review)]

Summary:

In this study, the authors performed a screening for PDXP inhibitors to identify compounds that could increase levels of pyridoxal 5'- phosphate (PLP), the co-enzymatically active form of vitamin B6. For the screening of inhibitors, they first evaluated a library of about 42,000 compounds for activators and inhibitors of PDXP and secondly, they validated the inhibitor compounds with a counter-screening against PGP, a close PDXP relative. The final narrowing down to 7,8-DHF was done using PLP as a substrate and confirmed the efficacy of this flavonoid as an inhibitor of PDXP function. Physiologically, the authors show that, by acutely treating isolated wild-type hippocampal neurons with 7,8-DHF they could detect an increase in the ratio of PLP/PL compared to control cultures. This effect was not seen in PDXP KO neurons.

Strengths:

The screening and validation of the PDXP inhibitors have been done very well because the authors have performed crystallographic analysis, a counter screening, and mutation analysis. This is very important because such rigor has not been applied to the original report of 7,8 DHF as an agonist for TrkB. Which is why there is so much controversy on this finding.

Weaknesses:

As mentioned in the summary report the study may benefit from some in vivo analysis of PLP levels following 7,8-DHF treatment, although I acknowledge that it may be challenging because of the working out of the dosage and timing of the procedure.

---

## [Referee Report · Reviewer #3 (Public Review)]

This is interesting biology. Vitamin B6 deficiency has been linked to cognitive impairment. It is not clear whether supplements are effective in restoring functional B6 levels. Vitamin B6 is composed of pyridoxal compounds and their phosphorylated forms, with pyridoxal 5-phosphate (PLP) being of particular importance. The levels of PLP are determined by the balance between pyridoxal kinase and phosphatase activities. The authors are testing the hypothesis that inhibition of pyridoxal phosphatase (PDXP) would arrest the age-dependent decline in PLP, offering an alternative therapeutic strategy to supplements. Published data illustrating that ablation of the Pdxp gene in mice led to increases in PLP levels and improvement in learning and memory trials are consistent with this hypothesis.

In this report, the authors conduct a screen of a library of ~40k small molecules and identify 7,8-dihydroxyflavone (DHF) as a candidate PDXP inhibitor. They present an initial characterization of this micromolar inhibitor, including a co-crystal structure of PDXP and 7,8-DHF. In addition, they demonstrate that treatment of cells with 7,8 DHP increases PLP levels. Overall, this study provides further validation of PDXP as a therapeutic target for the treatment of disorders associated with vitamin B6 deficiency and provides proof-of-concept for inhibition of the target with small-molecule drug candidates.

Strengths include the biological context, the focus on an interesting and under-studied class of protein phosphatases that includes several potential therapeutic targets, and the identification of a small molecule inhibitor that provides proof-of-concept for a new therapeutic strategy. Overall, the study has the potential to be an important development for the phosphatase field in general.

Weaknesses include the fact that the compound is very much an early-stage screening hit. It is an inhibitor with micromolar potency for which mechanisms of action other than inhibition of PDXP have been reported. Extensive further development will be required to demonstrate convincingly the extent to which its effects in cells are due to on-target inhibition of PDXP.

---

## [Author Response]

The following is the authors’ response to the original reviews.

**eLife assessment**
Following small molecule screens, this study provides convincing evidence that 7,8 dihydroxyflavone (DHF) is a competitive inhibitor of pyridoxal phosphatase. These results are important since they offer an alternative mechanism for the effects of 7,8 dihdroxyflavone in cognitive improvement in several mouse models. This paper is also significant due to the interest in the protein phosphatases and neurodegeneration fields.
**Public Reviews:**

**Reviewer #1 (Public Review):**
Summary:Zink et al set out to identify selective inhibitors of the pyridoxal phosphatase (PDXP). Previous studies had demonstrated improvements in cognition upon removal of PDXP, and here the authors reveal that this correlates with an increase in pyridoxal phosphate (PLP; PDXP substrate and an active coenzyme form of vitamin B6) with age. Since several pathologies are associated with decreased vitamin B6, the authors propose that PDXP is an attractive therapeutic target in the prevention/treatment of cognitive decline. Following high throughput and secondary small molecule screens, they identify two selective inhibitors. They follow up on 7, 8 dihydroxyflavone (DHF). Following structure-activity relationship and selectivity studies, the authors then solve a co-crystal structure of 7,8 DHF bound to the active site of PDXP, supporting a competitive mode of PDXP inhibition. Finally, they find that treating hippocampal neurons with 7,8 DHF increases PLP levels in a WT but not PDXP KO context. The authors note that 7,8 DHF has been used in numerous rodent neuropathology models to improve outcomes. 7, 8 DHF activity was previously attributed to activation of the receptor tyrosine kinase TrkB, although this appears to be controversial. The present study raises the possibility that it instead/also acts through modulation of PLP levels via PDXP, and is an important area for future work.Strengths:The strengths of the work are in the comprehensive, thorough, and unbiased nature of the analyses revealing the potential for therapeutic intervention in a number of pathologies.Weaknesses:Potential weaknesses include the poor solubility of 7,8 DHF that might limit its bioavailability given its relatively low potency (IC50 = 0.8 uM), which was not improved by SAR. However, the compound has an extended residence me and the co-crystal structure could aid the design of more potent molecules and would be of interest to those in the pharmaceutical industry. The images related to crystal structure could be improved.
**Reviewer #2 (Public Review):**
Summary:In this study, the authors performed a screening for PDXP inhibitors to identify compounds that could increase levels of pyridoxal 5'- phosphate (PLP), the co-enzymatically active form of vitamin B6. For the screening of inhibitors, they first evaluated a library of about 42,000 compounds for activators and inhibitors of PDXP and secondly, they validated the inhibitor compounds with a counter-screening against PGP, a close PDXP relative. The final narrowing down to 7,8-DHF was done using PLP as a substrate and confirmed the efficacy of this flavonoid as an inhibitor of PDXP function. Physiologically, the authors show that, by acutely treating isolated wild-type hippocampal neurons with 7,8-DHF they could detect an increase in the ratio of PLP/PL compared to control cultures. This effect was not seen in PDXP KO neurons.Strengths:The screening and validation of the PDXP inhibitors have been done very well because the authors have performed crystallographic analysis, a counter screening, and mutation analysis. This is very important because such rigor has not been applied to the original report of 7,8 DHF as an agonist for TrkB. Which is why there is so much controversy on this finding.Weaknesses:As mentioned in the summary report the study may benefit from some in vivo analysis of PLP levels following 7,8-DHF treatment, although I acknowledge that it may be challenging because of the working out of the dosage and timing of the procedure.
**Reviewer #3 (Public Review):**
This is interesting biology. Vitamin B6 deficiency has been linked to cognitive impairment. It is not clear whether supplements are effective in restoring functional B6 levels. Vitamin B6 is composed of pyridoxal compounds and their phosphorylated forms, with pyridoxal 5-phosphate (PLP) being of particular importance. The levels of PLP are determined by the balance between pyridoxal kinase and phosphatase activities. The authors are testing the hypothesis that inhibition of pyridoxal phosphatase (PDXP) would arrest the age-dependent decline in PLP, offering an alternative therapeutic strategy to supplements. Published data illustrating that ablation of the Pdxp gene in mice led to increases in PLP levels and improvement in learning and memory trials are consistent with this hypothesis.In this report, the authors conduct a screen of a library of ~40k small molecules and identify 7,8dihydroxyflavone (DHF) as a candidate PDXP inhibitor. They present an initial characterization of this micromolar inhibitor, including a co-crystal structure of PDXP and 7,8-DHF. In addition, they demonstrate that treatment of cells with 7,8 DHP increases PLP levels. Overall, this study provides further validation of PDXP as a therapeutic target for the treatment of disorders associated with vitamin B6 deficiency and provides proof-of-concept for inhibition of the target with small-molecule drug candidates.Strengths include the biological context, the focus on an interesting and under-studied class of protein phosphatases that includes several potential therapeutic targets, and the identification of a small molecule inhibitor that provides proof-of-concept for a new therapeutic strategy. Overall, the study has the potential to be an important development for the phosphatase field in general.Weaknesses include the fact that the compound is very much an early-stage screening hit. It is an inhibitor with micromolar potency for which mechanisms of action other than inhibition of PDXP have been reported. Extensive further development will be required to demonstrate convincingly the extent to which its effects in cells are due to on-target inhibition of PDXP.
**Recommendations for the authors:**
There is general agreement that the study represents an advance regarding the mechanisms of pyridoxal phosphatase and 7,8 DHF. From the reviewers' comments, several major questions and considerations are raised, followed by their detailed remarks:(1) More analysis of the solubility and dose of 7,8 DHF with regard to the 50% inhibition and the salt bridge of the B protomer, as raised by the reviewers.(2) Is there a possible involvement of another phosphatase?(3) Does 7,8 DHF cause an effect upon TrkB tyrosine phosphorylation?

We thank the Reviewers and Editors for their fair and constructive comments and suggestions. We have performed additional experiments to address these questions and considerations. In addition, we have generated two new high-resolution (1.5 Å) crystal structures of human PDXP in complex with 7,8-DHF that substantially expand our understanding of 7,8-DHF-mediated PDXP inhibition. The scientist who performed this work for the revision of our manuscript has been added as an author (shared first authorship).

We believe that the insights gained from these new data have further strengthened and improved the quality of our manuscript. Together, our data provide compelling evidence that 7,8-dihydroxyflavone is a direct and competitive inhibitor of pyridoxal phosphatase.

Please find our point-by-point responses to the Public Reviews that are not addressed in the Recommendations for the Authors, and the Recommendations for the Authors below.

**Reviewer #2:**
As mentioned in the summary report the study may benefit from some in vivo analysis of PLP levels following 7,8-DHF treatment, although I acknowledge that it may be challenging because of the working out of the dosage and timing of the procedure.

We agree that an in vivo analysis of PLP levels following 7,8-DHF treatment could be informative for the further evaluation of a possible mechanistic link between the reported effects of this compound and PDXP/vitamin B6. However, we currently do not have a corresponding animal experimentation permission in place and are unlikely to obtain such a permit within a reasonable me frame for this revision.

**Recommendations For The Authors:**

**Reviewer #1:**
The work is already well-written, comprehensive, and convincing.Suggestions that could improve the manuscript.(1) Include a protein tyrosine phosphatase (PTP) in the selectivity analysis. One possibility is that 7,8 DHF acts on a PTP (such as PTP1B), leading to TrkB activation by preventing dephosphorylation. I note that a previous study has looked at SAR for flavones with PTP1B (PMID: 29175190), which is worth discussion.

We thank the reviewer for bringing this interesting possibility to our attention. We were not aware of the SAR study for flavonoids with PTP1B by Proenca et al. but have now tested the effect of 7,8-DHF on PTP1B, referring to this paper. As shown in Figure 2d, PTP1B was not inhibited by 7,8-DHF at a concentration of 5 or 10 µM. At the highest tested concentration of 40 µM, 7,8-DHF inhibited PTP1B merely by ~20%. For comparison, compound C13 (3-hydroxy-7,8-dihydroxybenzylflavone-3’,4’dihydroxymethyl-phenyl), which emerged as the most active flavonoid in the SAR study by Proenca et al. inhibited PTP1B with an IC50 of 10 µM. Consistent with the results of these authors, our finding confirms that less polar substituents, such as O-benzyl groups at positions 7 and 8, and O-methyl groups at positions 3’ and 4’ of the flavone scaffold, are important for the ability of flavonoids to effectively inhibit PTP1B. We conclude that PTP1B inhibition by 7,8-DHF is unlikely to be a primary contributor to the reported cellular and in vivo effects of this flavone.

In addition to PTP1B, we have now additionally tested the effect of 7,8-DHF on the serine/threonine protein phosphatase calcineurin/PP2B, the DNA/RNA-directed alkaline phosphatase CIP, and three other metabolite-directed HAD phosphatases, namely NANP, NT5C1A and PNKP. PP2B, CIP and NANP were not inhibited by 7,8-DHF. Similar to PTP1B, PNKP activity was attenuated (~30%) only at 40 µM 7,8-DHF. In contrast, 7,8-DHF effectively inhibited NT5C1A (IC50 ~10 µM). NT5C1A is an AMP hydrolase expressed in skeletal muscle and heart. To our knowledge, a role of NT5C1A in the brain has not been reported. Based on currently available information, the inhibition of NT5C1A therefore appears unlikely to contribute to 7,8-DHF effects in the brain.

The results of these experiments are shown in the revised Figure 2d. Taken together, the extended selectivity analysis of 7,8-DHF on a total of 12 structurally and functionally diverse protein- and nonprotein-directed phosphatases supports our initial conclusion that 7,8-DHF preferentially inhibits PDXP.

(2) Line 144: It is unclear how fig 2c supports the statement here. Remove call out for clarity.

Our intention was to highlight the fact that 7,8-DHF concentrations >12.5 µM could not be tested in the BLI assay (shown in Figure 2c) due to 7,8-DHF solubility issues under these experimental conditions. However, since this is discussed in the text, but not directly visible in Figure 2c, we agree with the Reviewer and have removed this call out.

(3) Figure 3a. It is difficult to see the pink 7,8 DHF on top of the pink ribbon backbone. A better combination of colours could be used. Likewise in Figure 3b it is pink on pink again.

We have improved the combination of colors to enhance the visibility of 7,8-DHF and have consistently color-coded murine and the new human PDXP structures throughout the manuscript.

(4) Figure 3c and d. These are the two protomers I believe, but the colour coding is not present in 3c where the ribbon is now gray. Please choose colours that can be used to encode protomers throughout the figure.

Please see response to point 3 above.

(5) Figure 3f. I think this is the same protomer as 3c but a 180-degree rotation. Could this be indicated, or somehow lined up between the two figures for clarity? It would also be useful to have 3e in the same orientation as 3f, to better visualise the overlap with PLP binding. PLP and 7,8 DHF could be labelled similarly to the amino acids in 3f (the colour coding here is helpful).

Please see response to point 3 above. We have substantially revised the structural figures and have used consistent color coding and the same perspective of 7,8-DHF in the PDXP active sites.

(6) Figure 3g. The colours of the bars relating to specific mutations do not quite match the colours in Figure 3f, which I think was the aim and is very helpful.

We have adapted the colours of the residues in Figure 3f (now Fig. 3b and additionally Fig. 3 – figure supplement 1e) so that they exactly match the colours of the bars in Figure 3g (now Fig. 3d).

**Reviewer #2:**

No further comments.

**Reviewer #3:**
Page 4: The authors describe 7,8DHF as a "selective" inhibitor of PDXP - in my opinion, they do not have sufficient data to support such a strong assertion. Reports that 7,8DHF may act as a TRK-B-agonist already highlight a potential problem of off-target effects. Does 7,8DHF promote tyrosine phosphorylation of TRK-B in their hands? The selectivity panel presented in Figure 2, focusing on 5 other HAD phosphatases, is much too limited to support assertions of selectivity.

We agree with the Reviewer that our previous selectivity analysis with six HAD phosphatases was limited. To further explore the phosphatase target spectrum of 7,8-DHF, we have now analyzed six other enzymes: three other non-HAD phosphatases (the tyrosine phosphatase PTP1B, the serine/threonine protein phosphatase PP2B/calcineurin, and the DNA/RNA-directed alkaline phosphatase/CIP) and three other non-protein-directed C1/C0-type HAD phosphatases (NT5C1A, NANP, and PNKP). The C1-capped enzymes NT5C1A and NANP were chosen because we previously found them to be sensitive to small molecule inhibitors of the PDXP-related phosphoglycolate phosphatase PGP (PMID: 36369173). PNKP was chosen to increase the coverage of C0-capped HAD phosphatases (previously, only the C0-capped MDP1 was tested).

We found that calcineurin, CIP and NANP were not inhibited by up to 40 µM 7,8-DHF. The activities of PTP1B or PNKP activity were attenuated (by ~20 or 30%, respectively) only at 40 µM 7,8-DHF. In contrast, 7,8-DHF effectively inhibited NT5C1A (IC50 ~10 µM). We have previously found that NT5C1A was sensitive to small-molecule inhibitors of the PDXP paralog PGP, although these molecules are structurally unrelated to 7,8-DHF (PMID: 36369173). NT5C1A is an AMP hydrolase expressed in skeletal muscle and heart (PMID: 12947102). To our knowledge, a role of NT5C1A in the brain has not been reported. Based on currently available information, the inhibition of NT5C1A therefore appears unlikely to contribute to 7,8-DHF effects in the brain. The results of these experiments are shown in the revised Figure 2d. Taken together, the extended selectivity analysis of 7,8-DHF on a total of 12 structurally and functionally diverse protein- and non-protein-directed phosphatases supports our initial conclusion that 7,8-DHF preferentially inhibits PDXP. To nevertheless avoid any overstatement, we have now also replaced “selective” by “preferential” in this context throughout the manuscript.

We have not tested if 7,8-DHF promotes tyrosine phosphorylation of TRK-B. Being able to detect 7,8-DHF-induced TRK-B phosphorylation in our hands would not exclude an additional role forPDXP/vitamin B6-dependent processes. Not being able to detect TRK-B phosphorylation may indicate absence of evidence or evidence of absence. This would neither conclusively rule out a biological role for 7,8-DHF-induced TRK-B phosphorylation in vivo, nor contribute further insights into a possible involvement of vitamin B6-dependent processes in 7,8-DHF induced effects.

Page 6: The authors report that they obtained only two PDXP-selective inhibitor hits from their screen; 7,8DHF and something they describe as FMP-1. For the later, they state that it "was obtained from an academic donor, and its structure is undisclosed for intellectual property reasons". In my opinion, this is totally unacceptable. This is an academic research publication. If the authors wish to present data, they must do so in a manner that allows a reader to assess their significance; in the case of work with small molecules that includes the chemical structure. In my opinion, the authors should either describe the compound fully or remove mention of it altogether.

We are unable to describe “FMP-1” because its identity has not been disclosed to us. The academic donor of this molecule informed us that they were not able to permit release of any details of its structure or general structural class due to an emerging commercial interest.

We mentioned FMP-1 simply to highlight the fact that the screening campaign yielded more than one inhibitor. FMP-1 was also of interest due its complete inhibition of PDXP phosphatase activity.

Because the structure of this molecule is unknown to us, we have now removed any mention of this compound in the manuscript. For the same reason, we have removed the mention of the inhibitor hits “FMP-2” and “FMP-3” in Figure 2 – figure supplement 1 and Figure 2 – figure supplement 2. The number of PDXP inhibitor hits in the manuscript has been adapted accordingly.

Page 7: The observed plateau at 50% inhibition requires further explanation. It is not clear how poor solubility of the compound explains this observation. For example, the authors state that "due to the aforementioned poor solubility of 7,8DHF, concentrations higher than 12.5µM could not be evaluated". Yet on page 8, they describe assays against the specificity panel at concentrations of compound up to 40µM. Do the analogues of 7,8DHF (Fig 2b) result in >50% inhibition at higher concentrations? Further explanation and data on the solubility of the compounds would be of benefit.

We currently do not have a satisfactory explanation for the apparent plateau of ~50% PDXP inhibition by 7,8-DHF. Resolving this question will likely require other approaches, including computational chemistry such as molecular dynamics simulations, and we feel that this is beyond the scope of the present manuscript.

We previously speculated that the limited solubility of 7,8-DHF may counteract a complete enzyme inhibition if higher concentrations of this molecule are required. Specifically, we referred to Todd et al. who have performed HPLC-UV-based solubility assays of 7,8-DHF (35). These authors found that immediately after 7,8-DHF solubilization, nominal 7,8-DHF concentrations of 5, 20 or 50 µM resulted in 0.5, 3.0 or 13 µM of 7,8-DHF in solution of (i.e., 10, 15 or 26% of the respective nominal concentration). Seven hours later, 46, 26 or 26% of the respective nominal 7,8-DHF concentrations were found in solution. Hence, above a nominal concentration of 5 µM, 7,8-DHF solubility does not increase linearly with the input concentration, but plateaus at ~20% of the nominal concentration. This phenomenon could potentially contribute to the apparent plateau of human or murine PDXP inhibition by 7,8-DHF in vitro.

However, experiments performed during the revision of our manuscript show that they HADphosphatase NT5C1A can be effectively inhibited by 7,8-DHF with an IC50-value of 10 µM (see revised Fig. 2). Together with the fact that the activity of the PDXP-Asn61Ser variant can be completely inhibited by 7,8-DHF (see Fig. 3d), we conclude that the reason for the observed plateau of PDXP inhibition is likely to be primarily structural, with Asn61 impeding 7,8-DHF binding. We have therefore removed the mention of the limited solubility of 7,8-DHF here. On p.14, we now say: “These data also suggest that Asn61 contributes to the limited efficacy of 7,8-mediated PDXP inhibition in vitro.”

The solubility of 7,8-DHF is dependent on the specific assay and buffer conditions. In BLI experiments, interference patterns caused by binding of 7,8-DHF in solution to biotinylated PDXP immobilized on the biosensor surface are measured. In phosphatase selectivity assays, phosphatases are in solution, and the effect of 7,8-DHF on the phosphatase activity is measured via the quantification of free inorganic phosphate.

In BLI experiments, we observed that the sensorgrams obtained with the highest tested 7,8-DHF concentration (25 µM) showed the same curve shapes as the sensorgrams obtained with 12.5 µM 7,8-DHF. This contrasts with the expected steeper slope of the curves at 25 µM vs. 12.5 µM 7,8-DHF. The same behavior was observed for the reference sensors (i.e., the SSA sensors that were not loaded with PDXP, but incubated with 7,8-DHF at all employed concentrations for referencing against nonspecific binding of 7,8-DHF to the sensors). The sensorgrams at 25 µM 7,8-DHF were therefore not included in the analysis (this is now specified in the Materials and Methods BLI section on p.27). To clarify this point, we now state that “As a result of the poor solubility of the molecule, a saturation of the binding site was not experimentally accessible” (p.7).

In contrast, the phosphatase selectivity assays described on p.8 could be performed with nominal 7,8-DHF concentrations of up to 40 µM. Although the effective 7,8-DHF concentration in solution is expected to be lower (see ref. 35 and discussed above), the limited solubility of 7,8-DHF in phosphatase assays does not prevent the quantification of free inorganic phosphate. Nevertheless, we cannot exclude some interference with this absorbance-based assay (e.g., due to turbidity caused by insoluble compound). Indeed, 5,6-dihydroxyflavone and 5,6,7-trihydroxyflavone caused an apparent increase in PDXP activity at concentrations above 10 µM (see Figure 2b), which may be related to compound solubility issues. Alternatively, these flavones may activate PDXP at higher concentrations.

We have tested the 7,8-DHF analogue 3,7,8,4’-tetrahydroxyflavone at concentrations of 70 and 100 µM. At concentrations >100 µM, the DMSO concentration required for solubilizing the flavone interferes with PDXP activity. PDXP inhibition by 3,7,8,4’-tetrahydroxyflavone was slightly increased at 70 µM compared to 40 µM (by ~18%) but plateaued between 70 and 100 µM. These results are now mentioned in the text (p.7): “The efficacy of PDXP inhibition by 3,7,8,4’-tetrahydroxyflavone was not substantially increased at concentrations >40 µM (relative PDXP activity at 40 µM: 0.46 ± 0.05; at 70 µM: 0.38 ± 0.15; at 100 µM: 0.37 ± 0.09; data are mean values ± S.D. of n=6 experiments).”

Page 9: The authors report that PDXP crystallizes as a homodimer in which 7,8DHF is bound only to one protomer. Is the second protomer active? Does that contribute to the 50% inhibition plateau? If Arg62 is mutated to break the salt bridge, does inhibition go beyond 50%?

We have no way to measure the activity of the second, inhibitor-free protomer in murine PDXP. We know that PDXP functions as a constitutive homodimer, and based on our current understanding, both protomers are active. We have previously shown that the experimental monomerization of PDXP (upon introduction of two-point mutants in the dimerization interface) strongly reduces its phosphatase activity. Specifically, PDXP homodimerization is required for an inter-protomer interaction that mediates the proper positioning of the substrate specificity loop. Thus, homodimerization is necessary for effective substrate coordination and -dephosphorylation (PMID: 24338687).

In the murine structure, we observed that 7,8-DHF binding to the second subunit (the B-protomer) is prevented by a salt bridge between Arg62 and Asp14 of a symmetry-related A-protomer in the crystal lace (i.e., this is not a salt bridge between Arg62 in the B-protomer and Asp14 in the A-protomer of a PDXP homodimer). As suggested, we have nevertheless tested the potential role of this salt bridge for the sensitivity of the PDXP homodimer to 7,8-DHF.

The mutation of Arg62 is not suitable to answer this question, because this residue is involved in the coordination of 7,8-DHF (see Figure 3b), and the PDXP-Arg62Ala mutant is inhibitor resistant (see Figure 3d). We have therefore mutated Asp14, which is not involved in 7,8-DHF coordination. As shown in the new Figure 3 – figure supplement 1d, the 7,8-DHF-mediated inhibition of PDXPAsp14Ala again reached a plateau at ~50%. This result suggests that while an Arg62-Asp14 salt bridge is stabilized in the murine crystal, it is not a determinant of the active site accessibility of protomer B in solution.

To address this important question further, we have now also generated co-crystals of human PDXP bound to 7,8-DHF, and refined two structures to 1.5 Å. We found that in human PDXP, both protomers bind 7,8-DHF. These new, higher resolution data are now shown in the revised Figure 3 and its figure supplements, and we have moved the panels referring to the previously reported murine PDXP structure to the Figure 3 – figure supplement 1. Thus, both protomers of human PDXP, but only one protomer of murine PDXP bind 7,8-DHF in the crystal structure, yet the 7,8-DHFmediated inhibition of human and murine PDXP plateaus at ~50% under the phosphatase assay conditions (see Figure 2a). We conclude that 7,8-DHF binding efficiency in the PDXP crystal does not necessarily reflect its inhibitory efficiency in solution.

Taken together, these data indicate that the apparent partial inhibition of murine and human PDXP phosphatase activity by 7,8-DHF in our in vitro assays is not explained by an exclusive binding of 7,8DHF to just one protomer of the homodimer.

Page 10-12; Is it possible to generate a mutant form of PDXP in which activity is maintained but inhibition is attenuated - an inhibitor-resistant mutant form of PDXP? Can such a mutant be used to assess on-target vs off-target effects of 7,8DHF in cells?

This is an excellent point, and we agree with the Reviewer that such an approach would provide further evidence for cellular on-target activity of 7,8-DHF. Indeed, the verification of the PDXP-7,8DHF interaction sites has led to the generation of catalytically active, inhibitor-resistant PDXP mutants, such as Tyr146Ala and Glu148Ala (Fig. 3d). However, the biochemical analysis of such mutants in primary hippocampal neurons is a very difficult task.

Primary hippocampal neurons are derived from pooled, isolated hippocampi of mouse embryos and are subsequently differentiated for 21 days in vitro. The resulting cellular yield is typically low and variable, and the viability (and contamination of the respective cultures with e.g. glial cells) varies from batch to batch. Although such cell preparations are suitable for electrophysiological or immunocytochemical experiments, they are far from ideal for biochemical studies. A meaningful experiment would require the efficient expression of a catalytically active, but inhibitor-resistant PDXP-mutant in PDXP-KO neurons. In parallel, PDXP-KO cells reconstituted with PDXP-WT (at phosphatase activity levels comparable with the PDXP mutant cells) would be needed for comparison. Unfortunately, the generation of (a) sufficient numbers of (b) viable cells that (c) efficiently express (d) functionally comparable levels of PDXP-WT or -mutant for downstream analysis (PLP/PL-levels upon inhibitor treatment) is currently not possible for us.

Human iPSC-derived (hippocampal) spheroids are at present no alternative, due to the necessity of generating PDXP-KO lines first, and the difficulties with transfecting/transducing them. Such a system would require extensive validation. We have attempted to use SH-SY5Y cells (a metastatic neuroblastoma cell line), but PDXK expression in these cells is modest and they produce too little PLP. We therefore feel that this question is beyond the scope of our current study.